# Transcription-coupled and epigenome-encoded mechanisms direct H3K4 methylation

Satoyo Oya [1] ✉, Mayumi Takahashi[2], Kazuya Takashima[2], Tetsuji Kakutani [1,2] ✉ & Soichi Inagaki [1,3] ✉

Mono-, di-, and trimethylation of histone H3 lysine 4 (H3K4me1/2/3) are associated with transcription, yet it remains controversial whether H3K4me1/2/3 promote or result from transcription. Our previous characterizations of *Arabidopsis* H3K4 demethylases suggest roles for H3K4me1 in transcription. However, the control of H3K4me1 remains unexplored in *Arabidopsis*, in which no methyltransferase for H3K4me1 has been identified. Here, we identify three *Arabidopsis* methyltransferases that direct H3K4me1. Analyses of their genome-wide localization using ChIP-seq and machine learning reveal that one of the enzymes cooperates with the transcription machinery, while the other two are associated with specific histone modifications and DNA sequences. Importantly, these two types of localization patterns are also found for the other H3K4 methyltransferases in *Arabidopsis* and mice. These results suggest that H3K4me1/2/3 are established and maintained via interplay with transcription as well as inputs from other chromatin features, presumably enabling elaborate gene control.

Posttranslational modifications of histone residues shape the epigenome and affect gene expression[1]. Histone H3 lysine 4 methylation (H3K4me) is a conserved histone mark found in actively transcribed genes; however, the molecular basis and functional significance of the interrelationship between H3K4me and transcription remain elusive[2,3]. It has been proposed that H3K4me can be viewed as a memory of transcription because transcription directs H3K4 methylation[4–7]. On the other hand, in some experimental systems, H3K4me is deposited independent of transcription[8,9]. The underlying mechanisms that coordinate transcription-coupled and transcription-independent H3K4me are largely unexplored.

Another layer of H3K4me complexity is its occurrence in three states (mono-, di-, and trimethylation; me1, me2 and me3, respectively). Generally, H3K4me3 and H3K4me2 are found on the first several nucleosomes of genes, while H3K4me1 is found in downstream regions within the gene bodies[10–12]. Our previous genetic and genomic studies in *Arabidopsis thaliana* (hereafter *Arabidopsis*) have demonstrated key roles of the gene body H3K4me1 in various epigenetic phenomena. A putative H3K4me1 demethylase, LYSINE-SPECIFIC DEMETHYLASE1-LIKE2 (LDL2), mediates silencing of genes with the repressive histone mark H3K9me2 by removal of H3K4me1[13]. Another related demethylase, FLOWERING LOCUS D (FLD), downregulates transcriptional elongation and initiation by removing H3K4me1 in genes that have high levels of convergent overlapping transcription (i.e., antisense transcription)[14]. Although these findings regarding the H3K4 demethylases of *Arabidopsis* suggest a link of H3K4me1 with other chromatin features and transcription, the mechanisms of H3K4me1 control remain elusive, mainly because methyltransferase(s) for H3K4me1 has not been identified in plants.

In both animals and plants, two types of H3K4 methyltransferases are known: Set1-type and Trithorax/Trithorax-related (Trx/Trr)-type H3K4 methyltransferases[15]. However, yeasts have lost Trx/Trr-type

[1]Department of Biological Sciences, Graduate School of Science, The University of Tokyo, Tokyo, Japan. [2]National Institute of Genetics, Mishima Japan. [3]PRESTO, Japan Science and Technology Agency, Kawaguchi, Japan. ✉e-mail: o.satoyo@bs.s.u-tokyo.ac.jp; tkak@bs.s.u-tokyo.ac.jp; soinagak@bs.s.u-tokyo.ac.jp

H3K4 methyltransferases during evolution[16], and yeast Set1 is the sole H3K4 methyltransferase responsible for all H3K4me1/2/3. *Arabidopsis* has five Trx/Trr-type enzymes (Arabidopsis Trithorax (ATX) 1 to 5) and at least one Set1-type enzyme (Arabidopsis Trithorax Related (ATXR) 7) (Fig. 1a). Additionally, ATXR3, which has a domain structure characteristic of Set1-type enzymes but contains an atypical catalytic domain as H3K4 methyltransferase[17,18], can catalyze all three states of H3K4me1-3 in vitro[19]. Loss-of-function mutations in the *ATXR3* gene cause substantial H3K4me3 loss genome-wide, but H3K4me1 and H3K4me2 are largely unaffected[19]. In addition, triple loss-of-function of *ATX3*, *ATX4*, and *ATX5* leads to global decreases in H3K4me3 and H3K4me2 but not H3K4me1[20]. However, methyltransferase mutants with a genome-wide decrease of H3K4me1 levels have not been reported. The functions of *ATX1*, *ATX2*, and *ATXR7* genes have been characterized in the context of development, flowering regulation, and plant immunity[19-27]. For example, they were shown to redundantly inhibit flowering by activating the transcription of the flowering repressor *FLOWERING LOCUS C* (*FLC*) by increasing H3K4me1/2/3 levels within the *FLC* locus[21,24]. In the control of *FLC*, ATXR7 was shown to counteract FLD[24], which demethylates H3K4me1[14]. However, the genome-wide impacts of these putative H3K4 methyltransferases have not been elucidated.

Here, we explored the involvement of seven H3K4 methyltransferase genes in H3K4me1 as well as H3K4me2/3 by analyzing single and multiple mutants using chromatin immunoprecipitation sequencing (ChIP-seq). H3K4me1 levels are substantially decreased by the simultaneous loss of the *ATX1*, *ATX2*, and *ATXR7* genes. Our results clarify the division of labor among H3K4 methyltransferase genes and provide powerful genetic materials for elucidating the functions of each H3K4 methylation state. In addition, we analyzed the genomic localization of these enzymes. The subsequent application of machine learning algorithms revealed that the ATXR7 protein colocalizes with the transcription machinery, while ATX1 and ATX2 are associated with other chromatin modifications (epigenome) and specific DNA sequences (genome). These two types of localization patterns were also found by the reanalysis of other Set1- and Trx/Trr-type H3K4 methyltransferases, including those of animals. These findings lead us to a new perspective: some H3K4me marks may function as records of transcription, while other H3K4me marks may function as mediators of information encoded in the epigenome and/or genome.

## Results

### H3K4me1 levels are decreased by the triple loss of *ATX1/2/R7* genes

To identify enzyme(s) involved in H3K4me1, we examined mutants of six predicted H3K4 methyltransferases: ATX1 to 5 and ATXR7 (Fig. 1a). Among the six single mutants, *atx2* and *atxr7* showed relatively large decreases in H3K4me1 levels (Supplementary Fig. 1a). Interestingly, the *atx2* and *atxr7* mutants differed in the affected regions within genes; *atxr7* showed a marked decrease in H3K4me1 in the 3′ half of the gene bodies, while *atx2* showed a decrease over a broader region (Fig. 1b). Cluster analyses based on the intragenic H3K4me1 pattern in the *atx1 to 5* and *atxr7* mutants revealed that *atx1* and *atx2* formed one cluster, while *atx3*, *atx4*, and *atx5* formed another cluster (Supplementary Fig. 1b). These similarities in the effects on the H3K4me1 profile coincide with the similarity of the domain architecture of the corresponding proteins (Fig. 1a).

As the lack of a strong effect in each of the single mutants may reflect redundancy, we examined mutants that lose function of multiple ATX(R) genes. Based on the similarities of the protein domain structures and the effects on H3K4me1 patterns, we examined *atx1/ atx2* double mutants and found stronger effects on H3K4me1 than were observed in either of the single mutants (Fig. 1c, Supplementary Fig. 1c). Additionally, *atx1/atxr7* and *atx2/atxr7* showed stronger

H3K4me1 decreases than each single mutant (Supplementary Fig. 1c, Supplementary Data 1). The effect was still stronger in triple *atx1/atx2/ atxr7* mutants (hereafter referred to as *atx1/2/r7*) (Fig. 1c, d, Supplementary Fig. 2a, Supplementary Data 1). ChIP-seq replicates with spike-in control confirmed that *atx1*, *atx2*, and *atxr7* additively impact H3K4me1 (Supplementary Fig. 1d, Supplementary Data 2). Western blot analyses confirmed that triple *atx1/2/r7* mutation (with two different sets of T-DNA insertion alleles) resulted in the loss of H3K4me1 (Fig. 1e, Supplementary Fig. 2a), while H3K4me2 and H3K4me3 were largely unaffected (Fig. 1e). Consistent with the idea that ATX1/2/R7 mediates monomethylation of H3K4, ChIP-seq analysis targeting unmethylated H3K4 (H3K4me0) revealed that ATXR1/2/R7-marked genes showed increased signals of H3K4me0 in *atx1/2/r7* compared to wild type (Supplementary Fig. 2b). Taken together, these results demonstrate that ATX1, ATX2, and ATXR7 contribute to H3K4me1 in a partially redundant manner.

Consistent with previous reports[19,20,26], our Western blot and ChIP-seq analyses showed that H3K4me3 and H3K4me2 modifications are mainly mediated by ATXR3 and ATX3/4/5, respectively (Fig. 1d, e, Supplementary Fig. 2c–f). Collectively, the results showed that H3K4me1, H3K4me2, and H3K4me3 levels were specifically reduced in *atx1/2/r7*, *atx3/4/5*, and *atxr3*, respectively (Fig. 1d, e). The genes that were strongly affected in each of these mutants (colored red, yellow, or blue in Fig. 1d) showed high corresponding H3K4me levels in the wild type (e.g., ATX1/2/R7-marked genes showed high H3K4me1 levels relative to those of other genes) (Supplementary Fig. 1f–h), and had specific characteristics (Supplementary Fig. 1e). For example, ATX3/4/ 5-marked genes showed lower levels of expression compared to others (Supplementary Fig. 1e), in line with the body of research that suggests that H3K4me2 is a repressive mark in plants; H3K4me2 colocalizes with other repressive marks such as H3K27me3[28] and anti-correlates with transcription[29]. Hypo-H3K4me2 activates transcription in rice[29] and during regeneration of *Arabidopsis*[30]. Interestingly, the target genes of each ATX(R) group were mutually exclusive (Fig. 1f), suggesting that distinct mechanisms direct each of the H3K4 methyltransferases to distinct target genes.

### ATXR7 localization is associated with RNAP2, while ATX1 and ATX2 localization is associated with other chromatin modifications

To investigate the mechanisms that specify the targets of each ATX(R) protein that regulates H3K4me1, we determined the genome-wide localization of the ATX1, ATX2 and ATXR7 proteins using transgenic plants expressing FLAG-tagged proteins (Fig. 2a). ChIP-seq analyses revealed that ATX1 and ATX2 localized around transcription start sites (TSSs), typically in the range of −150 ~ +300 bp from the TSS. ATXR7 localized around transcription termination sites (TTSs) in the range of −200 ~ +200 bp from the TTS (Fig. 2b–d). Genes showing H3K4me1 loss in *atx1/2/r7* mutants tended to be bound by ATX1, ATX2 and ATXR7 (Supplementary Fig. 3), supporting the conclusion that these enzymes mediate H3K4me1 on chromatin.

To elucidate the chromatin-targeting mechanism(s) of these ATX1/2/R7 proteins, we screened for the determinants of ATX1/2/R7 localization by using a machine learning algorithm (random forest). Using chromatin and genomic features listed in Fig. 3a, the algorithm was trained to distinguish ATX1/2/R7-bound genes from unbound genes. After training, the random forest algorithm reported the 'importance (mean decrease in Gini)' of each feature as a relative score reflecting how informative the feature was for classification (Fig. 3a–c). In principle, the values of the features with high 'importance' scores can be positively (colocalize) or negatively (exclusive) correlated with the localization level of the corresponding ATX1/2/R7 protein. By taking all the features into account, the localization of each ATX1/2/R7 protein was well explained (Fig. 3d–f; note the high area under the curve (AUC) values).

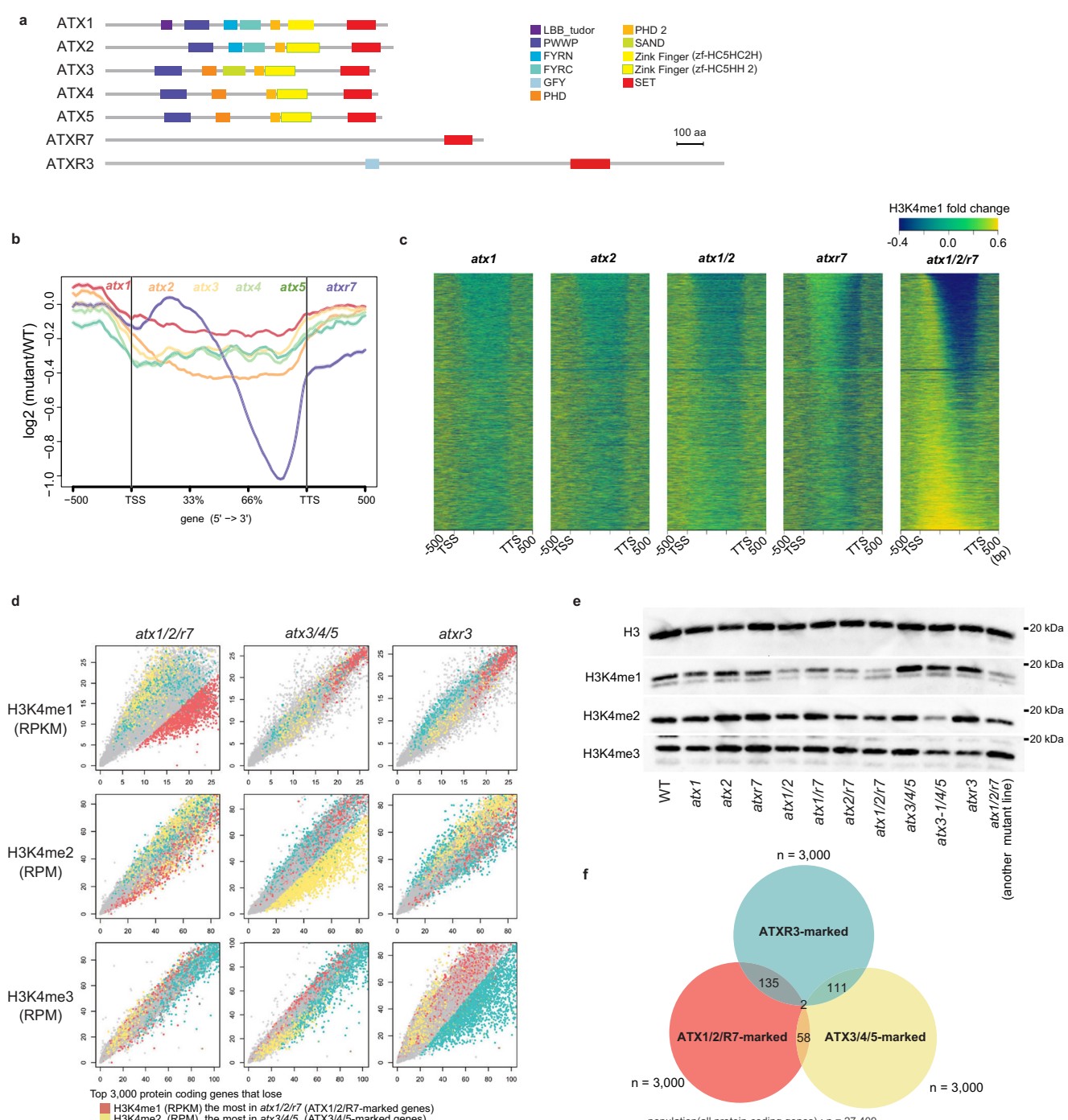

**Fig. 1 | *ATX1, ATX2,* and *ATXR7* redundantly contribute to H3K4me1. a** Domain architectures of ATX(R) proteins from the UniProt database. **b** Metaplot illustrates the averaged pattern of H3K4me1 changes of the 6 mutants compared to WT. **c** Mutations of *atx1, atx2,* and *atxr7* synergistically cause reduced H3K4me1 in the gene body. H3K4me1 changes from WT are visualized as heat maps. All genes are ordered so that genes that lost H3K4me1 the most in *atx1/2/r7* come to the top. **d** ChIP-seq for H3Kme1, H3K4me2, and H3K4me3 in mutants (*y*-axis) compared to WT (*x*-axis). Each dot represents each gene. Axes are trimmed to 0.98 quantiles. Top genes (*n* = 3000) that lose H3K4me1 most in *atx1/2/r7* (hereafter ATX1/2/R7-marked genes), lose H3K4me2 in *atx3/4/5* (ATX3/4/5-marked genes), and lose H3K4me3 in *atxr3* (ATXR3-marked genes) are colored red, yellow, and blue,

respectively. Values for H3K4me2/3 are RPM normalized instead of RPKM as in H3K4me1 because unlike H3K4me1, which covers the gene body, the amounts of H3K4me2 and H3K4me3 are less dependent on gene length[10]. **e** Western blotting of H3K4 methylations on bulk histone extracted from the mutants. The *atx1/2/r7* mutant on the right consists of three T-DNA alleles that are different from those in *atx1/2/r7* mutant mainly used in this study. Western blotting for H3K4me1 in *atx1/2/r7* were repeated and quantified (Supplementary Fig. 2a). **f** Activities of ATX1/2/R7, ATX3/4/5, and ATXR3 are observed in mutually exclusive genes. All pairwise ATX1/2/R7-marked genes, ATX3/4/5-marked genes, and ATXR3-marked genes have significantly fewer overlaps than expected (*n* = 328). Significances of the under-representation are tested by hypergeometric tests (*p* < 1e-4 for all pairs).

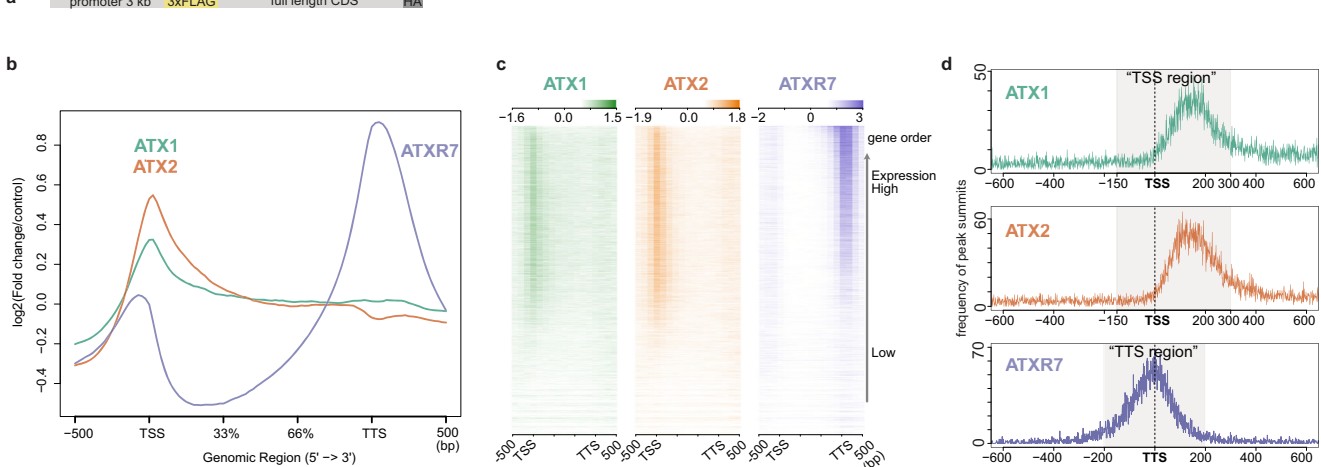

**Fig. 2 | ATX1 and ATX2 occupy TSS while ATXR7 occupies TTS. a** The transgene structure of tagged ATX(R) proteins used for ChIP-seq. **b, c** Metaplot (**b**) and heat maps (**c**) illustrating ATX1, ATX2, and ATXR7 distributions in the gene body region corrected with a non-transgenic control. The heat map was sorted so that highly transcribed genes (measured as mRNA-seq in WT) come to the top. **d** Position of ATX1 and ATX2 ChIP-seq peak summits relative to TSS and ATXR7 peaks relative to TTS (x-axis), visualized as a frequency of peak summits (y-axis), which are detected against non-transgenic control using MACS2 peak caller. The numbers of peaks are summarized in Supplementary Data 3. Most of the ATX1 and ATX2 peaks belong to the region spanning from 150 bp upstream to 300 bp downstream of TSS (hereafter 'TSS region'). Most of the ATXR7 peaks are between 200 bp upstream and 200 bp downstream of TTS (hereafter 'TTS region').

The most 'important' feature for distinguishing ATXR7-bound genes was the level of RNA Polymerase II (RNAP2) around the TTS (Fig. 3a). ATXR7-bound genes showed higher levels of total RNAP2 and RNAP2 phosphorylation at Ser2 and Ser5 sites in the carboxy-terminal domain (CTD) (Fig. 3g, Supplementary Fig. 4), indicating the strong colocalization of ATXR7 and RNAP2, especially when phosphorylated at Ser2 and Ser5. These results suggest that ATXR7, similarly to yeast Set1, may be recruited to chromatin in a transcription-coupled manner.

In contrast, the most 'important' feature for the prediction of ATX1- and ATX2-bound genes was H3K36me3 (Fig. 3b, c, Supplementary Fig. 5), which gives rise to three nonexclusive hypotheses: the distributions of ATX1/2 and H3K36me3 are driven by a shared factor; the presence of H3K36me3 drives ATX1/2 localization; or ATX1/2 drives H3K36me3 modifications. Consistent with the last hypothesis, *atx1/2/r7* induced the concomitant loss of H3K4me1 and H3K36me3 (Supplementary Fig. 6a), while H3K4me1 was unaffected by the *ash1 homolog 2* (*ashh2*) mutation, which causes a drastic reduction in H3K36me3 levels (Supplementary Fig. 6b)[31,32]. However, *ashh2* mutant keeps relatively high levels of H3K36me3 around TSS[33] where ATX1/2 localize (Fig. 2b–d), thus we cannot exclude the possibility that H3K36me3 acts upstream in addition to downstream of ATX1/2-H3K4me1. Among mutants for five H3K36 methyltransferase family genes, the *ashh3* mutant affects H3K36me3 at ATX1/2/R7-marked genes, while others including ASHH2, which has a H3K4me1-binding domain[34], markes genes that are not largely overlapped with ATX1/2/R7-marked genes (Supplementary Fig. 6c, d), consistent with the view that ASHH3 functions downstream of ATX1/2/R7-marked H3K4me1 to mediate H3K36me3.

In addition to H3K36me3, H2Bub and H4K16ac in TSS regions were indicated to be 'important' (Fig. 3b, c). ATX1- and ATX2-bound TSSs were rich in both modifications (Fig. 3h, i, Supplementary Fig. 4). Genetic depletion of H2Bub using *histone mono-ubiquitination 2 (hub2)* mutant resulted in decreased localization of ATX2 protein in genes that are predicted to lose ATX2 by in silico simulation of H2Bub loss (Supplementary Fig. 7a–c; for details of in silico simulation of H2Bub loss, see 'Random forest' section in Methods). Furthermore, those genes that become unbound by ATX2 in *hub2* mutant or by ATX1/2 in the in silico *hub* tend to lose H3K4me1 in *hub* mutants (*hub1* and *hub2*) (Supplementary Fig. 7d), consistent with the view that H2Bub promotes localization of ATX1/2, which then promotes H3K4me1.

Conversely, H2Bub was also decreased in H3K4me1-decreased genes in *atx1/2/r7* (Supplementary Fig. 7e).

These results indicate that H2Bub and H3K4me1 are mutually promoting each other. H4K16ac was not specifically affected at ATX1/2/R7-marked genes or ATX1/2-bound genes in *atx1/2/r7* (Supplementary Fig. 7e), suggesting that H4K16ac is not regulated downstream of ATX1/2.

On the other hand, the relative importance values of total or phosphorylated RNAP2 were markedly lower for ATX1/2 than for ATXR7 (Fig. 3a–c). These results suggest that ATX1/2 localization is governed by other chromatin modification(s), rather than by transcription. A previous study reported that physical interaction between ATX1 and RNAP2 phospho-Ser5 is involved in the recruitment of ATX1 to several ATX1-regulated genes[35]. Our results do not exclude the possibility that phospho-RNAP2 is also involved in the chromatin recruitment of ATX1/2 in addition to chromatin modification(s) such as H2Bub (Fig. 3, Supplementary Fig. 7).

The above results show that there are two modes of chromatin targeting. ATXR7 seems to cooperate with the transcriptional machinery, while ATX1 and ATX2 seem to target genes in a manner that is informed by chromatin modifications, regardless of the abundance of the transcriptional machinery. Interestingly, many genes occupied by ATXR7 are also bound by the H3K4me1 demethylase FLD[14], Supplementary Fig. 8a, b). In addition, genes showing the loss of H3K4me1 in the *atxr7* mutant tended to show a gain of H3K4me1 in the *fld* mutant (Supplementary Fig. 8c). Thus, ATXR7 and FLD localize in the TTS region and have opposite effects on H3K4me1.

## The DNA sequences of ATX1- and ATX2-bound TSSs have distinct architectures

Next, we asked whether the DNA sequences to which ATX1,2 or ATXR7 bind also have specific features. We converted the DNA sequences of TSS or TTS regions into a vector that represents the occurrence of all possible 6 base sequences (6-mer frequency vector) and trained the linear support vector machine (lSVM) algorithm to distinguish ATX1 (or ATX2)-bound TSS regions from unbound TSS regions, and ATXR7-bound TTS regions from unbound TTS regions. This process achieved good accuracy for ATX1 and ATX2 (Fig. 4a, b) but not for ATXR7 (Fig. 4c). The lSVM algorithm learns to assign greater weights to predictively important features. Consistent with their overlapping

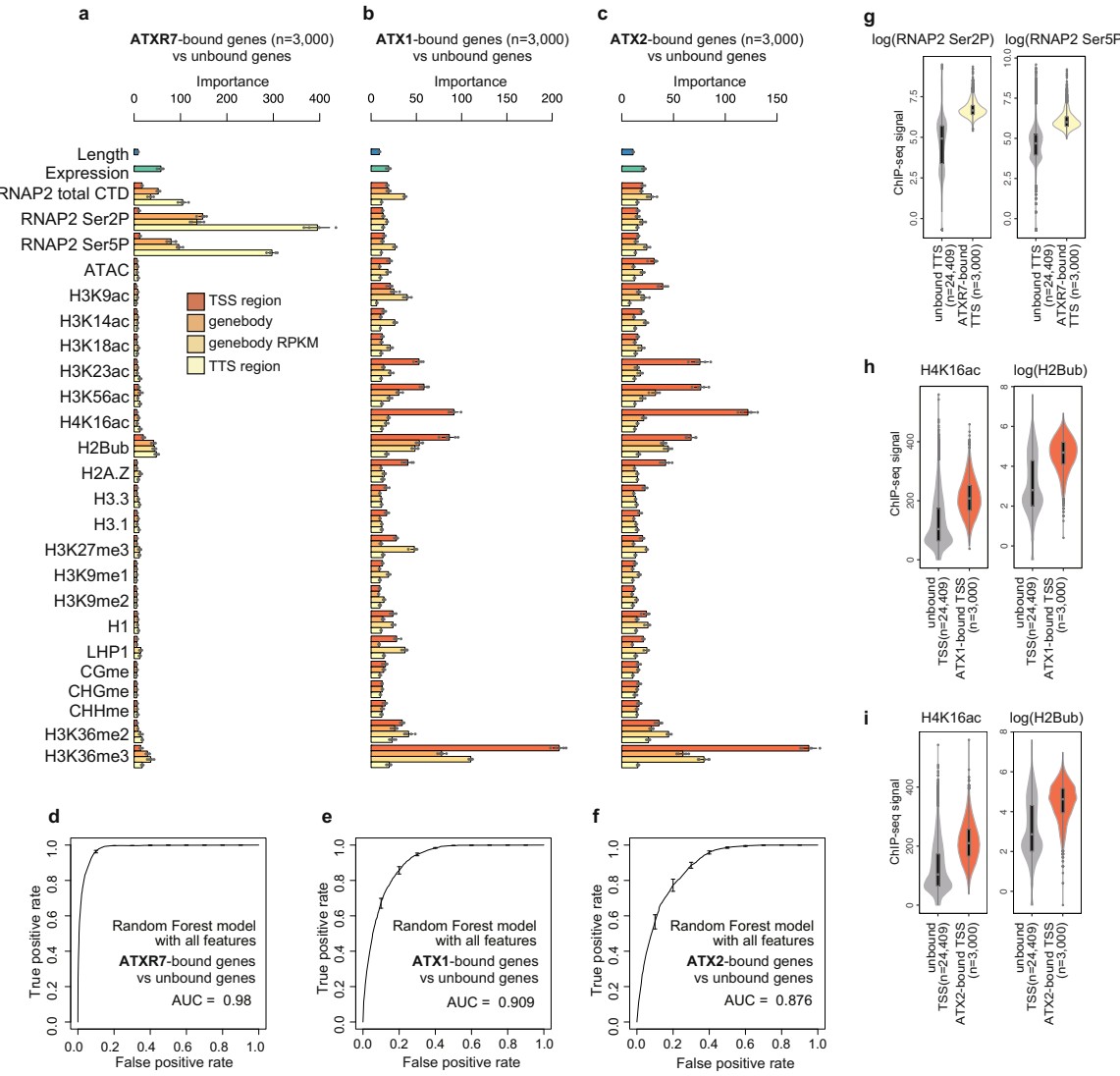

**Fig. 3 | ATXR7 localization is associated with RNAP2, while ATX1 and ATX2 localization is associated with other chromatin modifications. a–c** Chromatin features predictive of ATXR7 (**a**), ATX1 (**b**), and ATX2 (**c**) localization. ATXR7 (or ATX1,2)-bound genes were defined as the top 3000 genes with the highest ChIP-seq signal in 'TTS region (or TSS region)' compared to nontransgenic control. Random forest models were trained to predict ATXR7 (or ATX1,2)-bound and unbound genes. Bars indicate 'mean decrease in Gini' or in other words 'importance' derived from the random forest model, which is a relative score reflecting how informative the feature was for classification. 'Importance' are averaged from 5 repeats of training, each of which independently chose negative samples (unbound genes). Error bars are the standard deviation of the 5 repetitions. **d–f** ROC plots showing the prediction accuracy of the random forest models. AUC indicates the area under the ROC curve. ROC and AUC are calculated with data on Chr. 5, which were held out from training as test data. Average and standard deviation of the 5 repeats of training are plotted. **g–i** Violin plots showing the abundance of two most predictively 'important' features for each protein. Analyses with ChIP-seq datasets of biological replicates showed similar results (Supplementary Fig. 5). The center line of violin plot, median; box limits, upper and lower quartiles; whiskers, 1.5× interquartile range.

localization (Fig. 2c), the ATX1 and ATX2 models weighted similar sets of 6-mers (Fig. 4d), suggesting that ATX1 and ATX2 have similar mechanisms of sequence-based targeting.

To better understand the underlying mechanism, we sought to annotate the sets of DNA motifs that specify ATX1/2 binding. We selected the 120 predictive 6-mers, including the sixty 6-mers with the highest positive weights (when the TSS region contains these 6-mers, ATX1/2 are more likely to bind) and the sixty 6-mers with the greatest negative weights (opposing binding). The prediction accuracy for these 120 features was not much lower than that of the full model (Fig. 4a, b). Many of the 120 predictive 6-mers in ATX1/2 models were clustered into groups of overlapping 6-mer sequences (Fig. 4e, f, Supplementary Fig. 10a, b), in contrast to the situation for random or nonpredictive 6-mers (Supplementary Fig. 11a, b, 10c), suggesting the presence of >6 bp motifs that are predictive of ATX1/2 binding.

We annotated these discriminative DNA motifs by searching for matching sequences in the literature and in databases (Fig. 4e, f, Supplementary Fig. 10a, b, Supplementary Data 5, 6). In both the ATX1 and ATX2 models, TATA stretch was negatively weighted (Fig. 4e, Supplementary Fig. 10a, dotted circles). Consistent with this result, the occurrence of TATA stretch was less frequent in ATX-bound TSSs than in ATX-unbound TSSs (Fig. 4g, Supplementary Fig. 10d). On the other hand, GAGA stretch was positively weighted (Fig. 4f, Supplementary Fig. 10b, cyan circles), and ATX-bound TSSs showed broad GAGA peaks (Fig. 4g, Supplementary Fig. 10d). The positively weighted motifs also included 'ARGCC-CAWT' (in both the ATX1 and ATX2 models) and a telobox fragment (in the ATX1 model), the latter of which coincided with a trihelix transcription factor (TF) binding site (Fig. 4f, g and Supplementary Fig. 10b, d. Supplementary Fig. 12). RGCCCAW is likely bound by

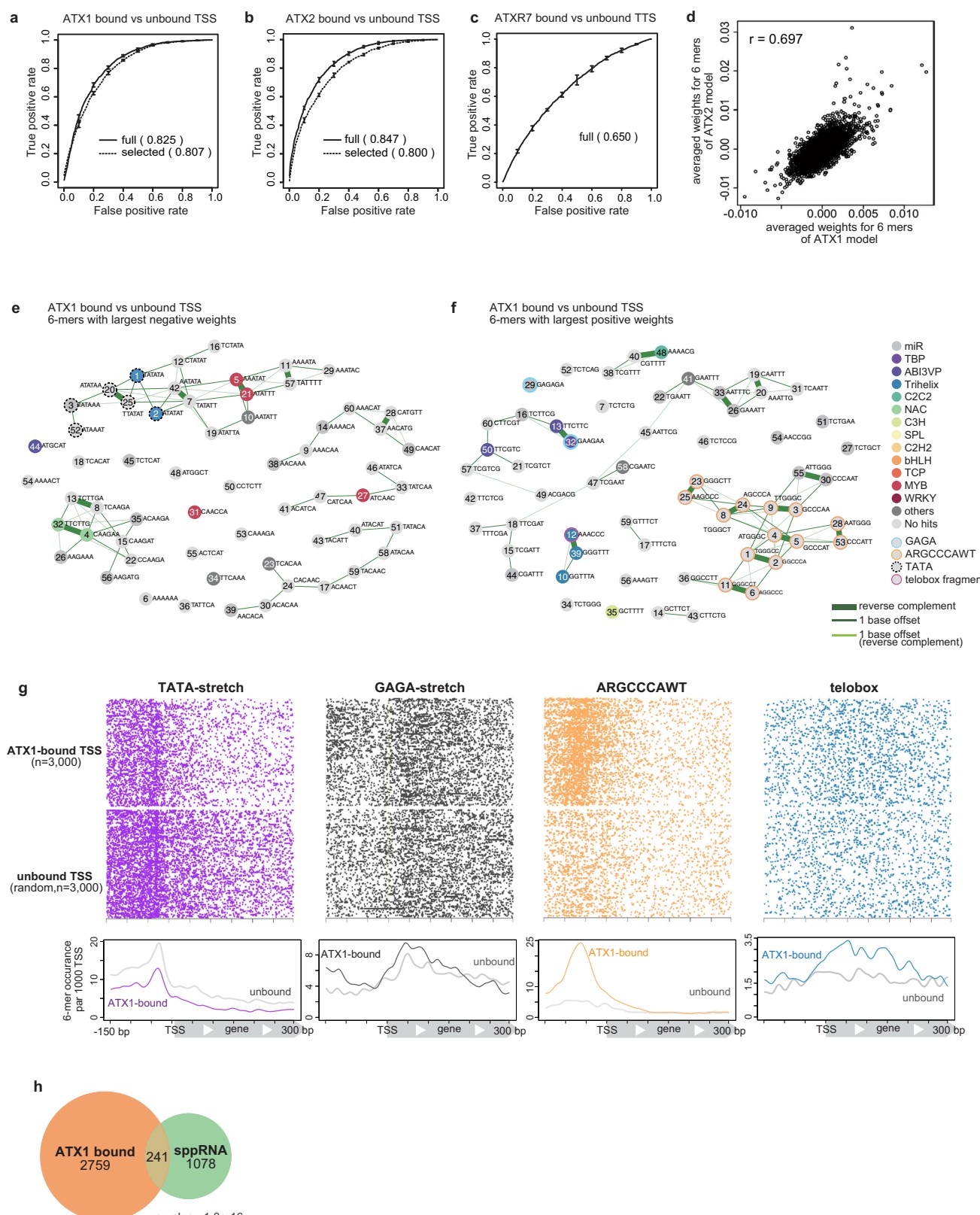

TCP family TFs[36–38]. While the ATX protein itself is not known to show DNA binding specificity, these additional DNA binders, such as the GAGA binding factor BPC6[39,40] and TCP and trihelix TFs, may physically link ATX to DNA motifs. The combination of GAGA and RGCCCAW is characteristic of TSSs that harbor short noncoding

RNAs (short promoter proximal RNAs, sppRNAs) with unknown functions[41]. Indeed, ATX1/2-bound genes were found to significantly overlap with sppRNA genes (Fig. 4h, Supplementary Fig. 10e), suggesting the existence of some links between sppRNA and ATX1/2 localization.

**Fig. 4 | The DNA sequences of ATX1- and ATX2-bound TSSs have distinct architectures. a–c** ROC plots that show predicting accuracy of linear SVM models, which are trained with the full set of 6-mers (solid line) and trained with the 120 6-mers which were highly weighted in the full model (dashed line), to discriminate (**a**) ATX1- or (**b**) ATX2-bound vs -unbound TSS DNA sequence, and (**c**) ATXR7-bound and -unbound TTS sequence. ROC and AUC are calculated with data on Chr 5, which are held out from the training as test data. Averaged scores of 5 cross-fold validation models are plotted. Error bars represent the standard deviation of the 5 repeats of training. Analyses with ChIP-seq datasets of biological replicates (see Methods) showed similar results (Supplementary Fig. 9). **d** Averaged SVM weights from ATX1 models (*x*-axis) correlates with those from ATX2 models (*y*-axis), indicating that similar sets of sequences predict ATX1 and ATX2 localizations. *r* = Pearson's correlation coefficient. **e, f** Clustering and annotation of predictive 6-mers. Each circle represents the top sixty 6-mers with negative (**e**) or positive (**f**) averaged SVM weights in the ATX1 models. Numbers within the circle are ranks of weights' absolute values (the more 'predictive' a 6-mer is, the smaller the number labels it). Pairs of related 6-mers are connected with lines; the thickest lines connect reverse complement pairs, the thinner lines connect neighboring 6-mers with 1 base offset, the thinnest lines connect reverse complement pairs with 1 base offset. Each 6-mers was searched for matching motifs (see Methods), then 6-mer circles were colored corresponding to the category of its top matched motif that meets q-value <0.1 criteria. 6-mers corresponding to ARGCCCAWT, telobox, GAGA, and TATA-stretch are manually highlighted with border circles. Some of the highly weighted motifs are evolutionarily conserved among TSS regions of the land plants, suggesting their functionality (Supplementary Fig. 13). **g** Positional distributions of the highly weighted motifs in the TSS region. **h** ATX1-bound TSS significantly overlaps with sppRNA-harboring TSS detected in the *hen2-2* background. The significance of the overlap was tested using a hypergeometric test.

These sequence-based learning results further support the idea that ATX1 and ATX2 target genes independent of transcription events. The predictive motif sets are further interpreted in the discussion.

## A positive correlation between transcription and H3K4me1 is mediated by ATXR7 and is disturbed by ATX1/2

From the above results we hypothesized that the ATX1 and ATX2 proteins monomethylate H3K4 irrespective of transcription, while ATXR7 does cotranscriptionally. We tested this hypothesis by analyzing the correlation between transcription and H3K4me1. In the wild type, the amount of H3K4me1 covering a gene is moderately correlated with the transcription level (see below). The hypothesis predicts that ATXR7 is responsible for this correlation through H3K4me1 deposition in transcribed genes, while ATX1 and ATX2 disrupt this correlation by depositing H3K4me1 regardless of transcription. In agreement with this prediction, the correlation became significantly stronger in *atx1/2* (Fig. 5a) compared to wild type (Fig. 5b), and weaker in *atxr7* (Fig. 5c), as quantified by Spearman's correlation coefficient or visualized by density heatmaps. This alteration in the correlation landscape was due to a change in H3K4me1 rather than transcription because the correlation showed the same trend when the transcription data for the wild type were used rather than those for the mutants (Supplementary Fig. 14a, b).

## Localization and functional mode of other *Arabidopsis* H3K4 methyltransferases

In addition to ATX1, ATX2, and ATXR7, *Arabidopsis* has at least four other putative H3K4 methyltransferases. Hence, we wondered how the localization patterns of the other methyltransferases were determined. A random forest analysis using the reported genome-wide distribution of the ATXR3 protein[42] revealed that the levels of total and phosphorylated RNAP2 and transcription presented the highest 'importance' and that they were significantly higher in ATXR3-bound genes than in ATXR3-unbound genes (Fig. 6a–d), suggesting a cotranscriptional mode of H3K4me3 deposition by ATXR3. Consistent with this interpretation, the strong correlation between transcript levels and H3K4me3 decreased in the *atxr3* mutant (Fig. 6e, f, Supplementary Fig. 15a, b).

The triple mutation of three other methyltransferases, ATX3/4/5, resulted in H3K4me2 loss; and the loss was also observed for H3K4me3 in the *atx3-1/4/5* mutant (for the details of two *atx3* alleles, see 'Plant materials' section in Methods) (Fig. 1d, e, Supplementary Fig. 2c, e, f, and ref. 20). The genome-wide distributions of the ATX3/4/5 proteins have not been reported. However, the correlation between H3K4me2 and transcription became more prominent in the absence of ATX3/4/5 (as observed for H3K4me3 in the *atx3-1/4/5* mutant) (Fig. 6g, h, Supplementary Fig. 15a, c–i), suggesting that these enzymes introduce H3K4me modifications in a transcription-independent manner, which could be instructed by other chromatin modifications or DNA sequences, as discussed for ATX1/2.

## Localization and functional mode of mammalian H3K4 methyltransferases

We wondered whether these two functional modes of H3K4 methyltransferases also occur in organisms other than plants. Figure 7a summarizes the lineage of H3K4 methyltransferases in eukaryotes. H3K4 methyltransferases are classified into Set1-type and Trx/Trr-type methyltransferases. ATXR7 and, probably, ATXR3 are Set1-type methyltransferases, while ATX1 to ATX5 belong to the Trx and/or Trr-type methyltransferases (Fig. 7a[18,43],). Fungi have lost Trx/Trr-type genes during evolution and exhibit only one Set1-type gene[16]. We analyzed mammalian SET1A, MLL2, and MLL3/4, for which genome-wide localization data in mouse ESCs are available. The reanalysis of the ChIP-seq datasets revealed that SET1A[44] and MLL2[45] localize around TSSs (typically in the range of −150-300 bp) and, to a lesser extent, around enhancers (Supplementary Fig. 16a, b). On the other hand, MLL3/4[46,47] localize to enhancers (typically in the range of −900-900 bp from the center of the enhancer) (Supplementary Fig. 16a, b).

According to the random forest model, the best predictor of both SET1A-bound TSSs and enhancers was the level of RNAP2 (Fig. 7b, c, Supplementary Fig. 16c, d). SET1A strongly colocalizes with RNAP2 (Fig. 7b, d). MLL2-bound TSSs were also best predicted by RNAP2 (Fig. 7e, f, Supplementary Fig. 16e, f). However, the prediction accuracy for the localization of MLL2 according to the RNAP2 level was lower than that for SET1A (Fig. 7d, g), and RNAP2 was not an outstanding predictor (Fig. 7e), suggesting that the colocalization of RNAP2-MLL2 was weaker than that of RNAP2-SET1A. The best predictor of MLL3/4-bound regions was not RNAP2 but H3K27ac, at both enhancers and TSSs (Fig. 7h, i, Supplementary Fig. 16g, h). At enhancers, which are the major localization sites of MLL3/4 (ref. 46,47, and Supplementary Fig. 16a, b), the observed localization was exclusive to sites of RNAP2 occurrence (Fig. 7h, j). These results suggest that SET1A is a cotranscriptional H3K4 methyltransferase, as is the case for its *Arabidopsis* homolog ATXR7 (and perhaps ATXR3), while MLL3/4 target chromatin irrespective of transcription, similarly to ATX1/2. MLL2 appears to be an intermediate. Therefore, it is plausible that Set1-type H3K4 methyltransferases tend to function in a cotranscriptional manner across eukaryotes, while Trx/Trr-type H3K4 methyltransferases are informed by other chromatin and DNA sequence features (Fig. 8).

## Discussion

Here, we revealed in *Arabidopsis* that the simultaneous loss of the SET-domain histone methyltransferase genes *ATX1*, *ATX2*, and *ATXR7* causes substantial H3K4me1 loss. In addition, consistent with previous studies[19,20,26], our results showed that ATX3, ATX4, and ATX5 redundantly regulate H3K4me2/3 and that ATXR3 regulates H3K4me3. By using a weak allele of ATX3, we were also able to obtain a mutant with

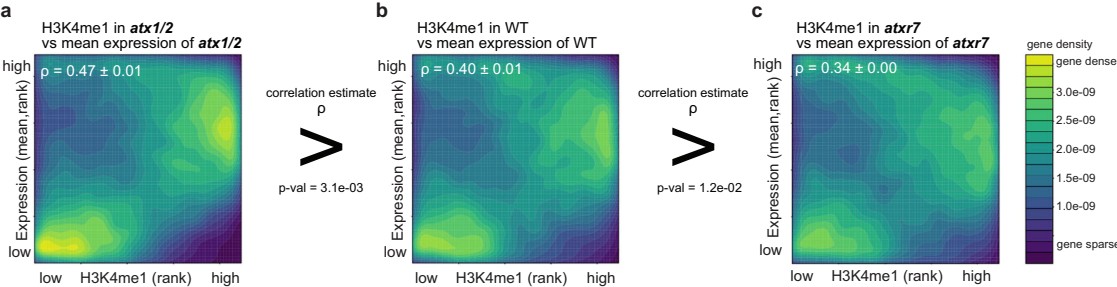

**Fig. 5 | Positive correlation between transcription and H3K4me1 is mediated by ATXR7 and is disturbed by ATX1/2. a–c** Protein-coding genes are ranked in order of H3K4me1 (x-axis, RPKM) and expression levels (y-axis, mRNA-seq FPKM, mean of the three RNA-seq replicates). The densities of genes are visualized as heat maps. ρ is the Spearman's correlation coefficient. Three replicates of mRNA-seq resulted in three ρ for each genotype, and the average and standard deviation of ρ are presented. P values indicate that correlation estimates ρ significantly differ between genotypes (Welch's two sample t-test). Another ChIP-seq datasets replicated consistent trends (Supplementary Fig. 14c–h).

specific H3K4me2 loss. Interestingly, these three groups of SET-domain genes are responsible for H3K4me1, H3K4me2/3, and H3K4me3 modifications among a distinct spectrum of genes. The genes that are affected by the loss of these three groups are mutually exclusive, and mutations within them affect H3K4me independently. For example, triple atx1/2/r7 mutation causes the loss of only H3K4me1 and not H3K4me2/3 on a set of genes (ATX1/2/R7-marked genes). This observation implies that other H3K4 methyltransferases can catalyze H3K4me2/3 modifications on unmethylated H3K4 (H3K4me0) in their target regions. Accordingly, ATXR3, for example, was shown to catalyze the H3K4me3 from H3K4me0 in vitro[19]. In addition, the observation that H3K4me2/3 of ATX1/2/R7-marked genes being not largely affected by atxr3 or atx3/4/5 mutation implies that multiple methyltransferases from the ATX1/2/R7, ATX3/4/5 and ATXR3 groups (and/or other unidentified H3K4 methyltransferase(s)) are redundantly involved in their deposition. Our study clarified the functional division of labor among ATX(R)s genome wide. In contrast to Arabidopsis, yeasts have a single H3K4 methyltransferase, Set1, which is responsible for all H3K4me modifications. The Arabidopsis mutants showing the specific reduction of each H3K4me state identified here will serve as powerful materials for understanding the specific functions of H3K4me1/2/3.

Previous works have reported that atx1 shows H3K4me3 loss[21–23,48,49] and that atx2 shows H3K4me2 loss[22], on the basis of ChIP-qPCR analyses of selected loci, while our data show that the redundant roles of these enzymes have the largest effect on the global H3K4me1 level. The catalytic domains of ATX1/2/R7 contain a bulky tyrosine at the tyrosine/phenylalanine switch (Supplementary Fig. 17), which is proposed to act as an obstacle to higher-order methylation[50], consistent with our conclusion that ATX1/2/R7 primarily regulates H3K4me1. The results of previous studies might reflect indirect effects triggered by altered transcription and/or minor locus-specific effects. Our conclusion is also supported by the results showing that ATX1/2/R7 redundantly repress flowering via FLC activation and that ATXR7 and FLD counteract each other in FLC regulation, probably by modulating H3K4me1[14,24,51].

Random forest analyses of the genome-wide localization of ATX1 and ATX2 suggested that H3K36me3, H4K16ac and H2Bub are candidate chromatin features for the recruitment of ATX1/2 (Fig. 3, Supplementary Fig. 4). Further examination clarified H2Bub indeed promotes localization of ATX1/2 (Supplementary Fig. 7), while H3K36me3 is promoted by H3K4me1 (Supplementary Fig. 6). The contributions of H4K16ac and H2Bub to ATX1/2 recruitment agree well with previous reports in other species. MLL3 and MLL4 (mammalian homologs of ATX) directly bind to H4K16ac via plant homeodomains (PHDs)[52]. ATX1 and ATX2 (but not ATXR7) also have PHD domains (Fig. 1a). H2Bub has long been known to promote H3K4 methyltransferase in yeasts and mammals and various explanatory

mechanisms have been proposed, while in plants ATXR3 was found not to follow the rule[42]. Recently, cryo-EM studies revealed that MLL-type H3K4 methyltransferase-containing complexes (complex proteins associated with Set1, COMPASS) attach more firmly to ubiquitinated nucleosomes[53].

Motif mining with the lSVM algorithm revealed three notable characteristics of the ATX1/2-bound TSS sequences that coincided with known features of animal Trx/Trr-bound sequences. First, the ATX1/2-bound regions are GAGA-type promoters rather than TATA-box promoters. In Arabidopsis, promoters without TATA boxes (a major core promoter motif) tend to contain GAGA stretch, which are suggested to be equivalent to animal CpG islands[54]. MLL2 also binds to non-TATA(= CpG)-type promoters[45]. Second, ATX1/2-bound TSS regions potentially recruit polycomb-group (PcG) proteins, considering that the combination of telobox and GAGA sequences is necessary and sufficient for the recruitment of Arabidopsis PcGs[55], which are antagonists of trithorax group (TrxG) proteins. In both plants and animals, TrxG and PcG often target the same genomic regions. Third, both ATX1/2-bound and MLL3/4-bound regions produce noncoding RNAs: sppRNAs and enhancer RNAs[46]. These results suggest that Trx/Trr methyltransferases bind to a common set of features across eukaryotes.

Our analyses suggested that Set1-type methyltransferases typically function in a cotranscriptional manner and that Trx/Trr-type methyltransferases are less informed by transcription. Although this bifurcation has not been explicitly noted previously, it may underlie the observations of a number of previous works. For instance, the concept that H3K4 methylation occurs cotranscriptionally was founded on studies of Set1 in budding yeast[4–7]. An interactome analysis of RNAP2 recovered Set1-type methyltransferases but not Trx/Trr-type methyltransferases in human cells[56], and the colocalization of RNAP2 with Set1-type, but not Trr/Trx-type methyltransferases, is prominent at the microscopic level in the Drosophila polytene chromosome[57]. In contrast, various recruitment factors in addition to RNAP2 have been proposed for Trx/Trr-type methyltransferases, including chromatin modifications as described above, TREs, and long noncoding RNAs[58]. These genome- and/or epigenome-informed methyltransferases may have provided the foundation for multicellularity via the elaborate control of H3K4me, considering that Trr/Trx-type methyltransferases are present in animals and plants but are absent in yeasts. Regulation by both transcription-coupled and (epi)genome-encoded mechanisms may be further generalized to other chromatin modifications, such as H3K36me[59].

In addition to transcription-coupled and (epi)genome-encoded pathways which regulate H3K4 methyltransferases, demethylases also shape the pattern of H3K4me. In Arabidopsis, the histone demethylase LDL2 removes H3K4me1 marks from the gene bodies that accumulate H3K9me2 and silences gene expression[13]. Another

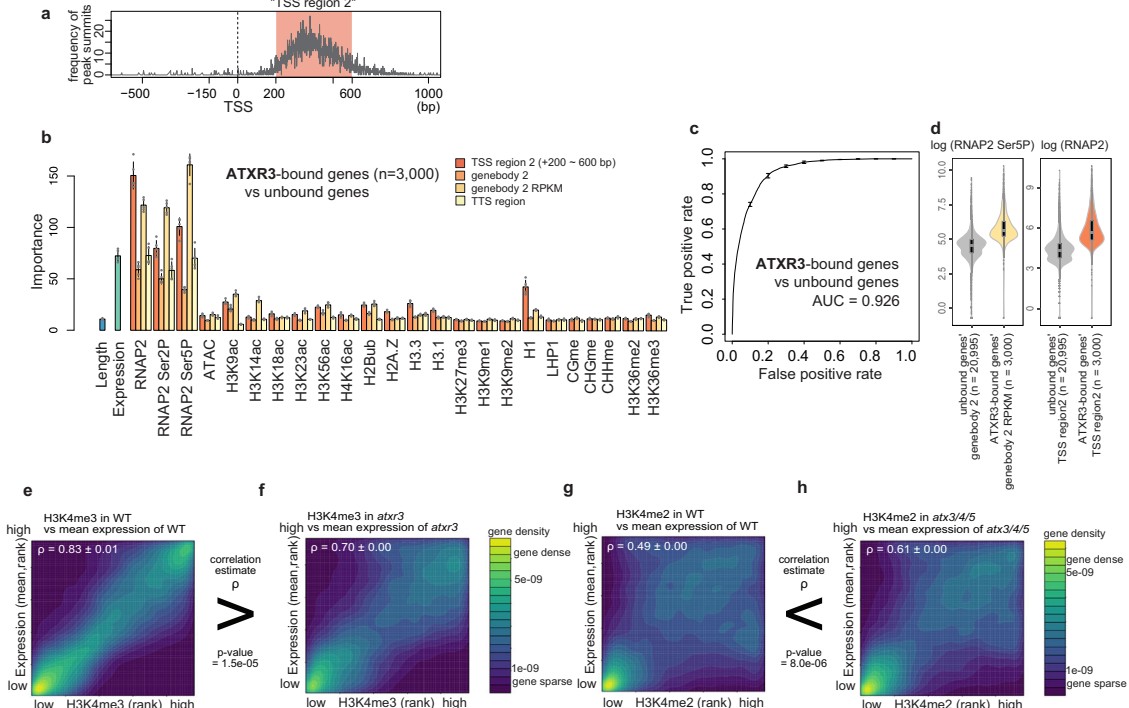

**Fig. 6 | Localization and functional mode of other Arabidopsis H3K4 methyltransferases. a** Position of ATXR3 peaks relative to TSS, visualized as a frequency of ChIP-seq peak summits around TSS. *x*-axis, distance from TSS; *y*-axis, number of peak summits. Most of the ATXR3 peaks belong to the region spanning from 200 bp to 600 bp downstream of TSS, which we hereafter refer to as 'TSS region 2'. **b** Chromatin features predictive of ATXR3 localization. Error bars represent the standard deviation of the 5 repeats of training. **c** ROC plot showing the prediction accuracy of the random forest models. AUC indicates the area under the ROC curve. ROC and AUC are calculated with data on Chr 5, which are held out from the training as test data. Average and standard deviation of the 5 repeats of training are plotted. **d** Violin plots showing the abundance of the two most predictively 'important' features. The center line of violin plot, median; box limits, upper and lower quartiles; whiskers, 1.5× interquartile range. **e–h** Protein-coding genes are ranked in order of H3K4me3 (**e**, **f**) or H3K4me2 (**g**, **h**) (*x*-axis, RPM) and expression levels (*y*-axis, mRNA-seq FPKM, mean of the three RNA-seq replicates). The densities of genes are visualized as heat maps. ρ is the Spearman's correlation coefficient. Three replicates of mRNA-seq resulted in three ρ for each genotype, and the average and standard deviation of ρ are represented. ChIP-seq data used is the data sets shown in Fig. 1d. *P* values indicate that correlation estimates ρ significantly differ between genotypes (Welch's two sample t-test).

histone demethylase, FLD, removes H3K4me1 from sites where convergent overlapping transcription takes place[14]. Although the biological function of H3K4me remains unclarified, these diverse mechanisms converging on H3K4me1 suggest that the roles of H3K4me require the coordination and integration of a wide range of information. H3K4me is proposed to coordinate splicing[60], transcriptional stability[61], cryptic transcription[62], sense-antisense transcriptions[14,63] and to confer transcriptional memory[64,65]. These apparent multifaceted functions may reflect the multiple regulation mechanisms of this modification. In future studies, the dissection of H3K4me based on its controlling mechanisms may help elucidate the functions of H3K4me.

## Methods

### Plant materials
T-DNA insertion mutants[66,67] used in this study were previously described. Names of mutant alleles such as *"atx1-1"* or *"atx1-2"* collide between papers, and the names provided below are not consensus, but examples; *atx1-2* (*sdg27*, SALK_149002C), *atx1-3* (SALK_119016C), *atx2-1* (*sdg30*, SALK_074806C), *atx2-2* (SALK_117262), *atx3-2* (*sdg14*, SAIL_582_H12, UTR insertion), *atx3-1* (*sdg14*, GK-128H01), *atx4* (*sdg16*, SALK_060156), *atx5* (*sdg29*, SAIL_705_H05), *atxr7-1* (*sdg25*, SALK_149692C), *atxr7-2* (SAIL_446_F12), *atxr3* (*sdg2*, SALK_021008), *ashh1* (*sdg26*, SALK_013895), *ashh2* (*sdg8*, SALK_065480), *ashh3* (*sdg7*, SALK_131218C), *ashh4* (*sdg24*, SK22803), *ashr3* (*sdg4*, SALK_128444), *hub1-4* (SALK_122512) and *hub2-2* (SALK_071289 C). For *atx1*, *atx2*, and *atxr7* mutants, *atx1-2*, *atx2-1*, and *atxr7-1* alleles were mainly used. *atx1-3, atx2-2, atxr7-2* were used as a replicate to

confirm bulk H3K4 methylation levels of *atx1/2/r7* mutant in Fig. 1e. For *atx3* mutants, we used SAIL_582_H12 (UTR insertion) if not stated otherwise. *atx3-1* allele (exon insertion[20]) was used in Fig. 1e and Supplementary Fig. 2f to compare with preceding research[20]. For tagged ATX1, ATX2, and ATXR7 lines, the constructs were made with the uniform design; native promoter, which is 3 kb region upstream from initiation codon, followed by 3 x FLAG sequence, coding regions (genomic DNA sequence including introns until just before stop codon) and HA sequence, cloned into pPLV01 vector[68]. The plasmids were transferred into respective single mutants (ATX1-tag into *atx1-2*, ATX2-tag into *atx2-1*, and ATXR7-tag into *atxr7-1*) via *Agrobacterium tumefaciens* GV3101::pMP90. Plant lines having homozygous transgene were selected in later generations. All the mutants and 'wild type (WT)' are in the Columbia-0 background. For all the experiments, seeds were sown on Murashige and Skoog (MS) plates and kept in dark at 4 °C for a few days, then grown for 15 days under long-day conditions (8 h dark and 16 h light) at 22 °C. Whole seedlings were used for experiments.

### Western blotting
Bulk histones for western blotting were prepared using Histone extraction kit (Active Motif) or essentially as described in ref. 69 briefly, 200 mg of frozen seedlings were disrupted into fine powder and suspended in 5 ml of nuclei isolation buffer (NIB: 10 mM MES-KOH pH 5.3, 10 mM NaCl, 10 mM KCl, 250 mM sucrose, 2.5 mM EDTA, 2.5 mM ß-mercaptoethanol, 0.1 mM spermine, 0.1 mM spermidine, 0.3% Triton X-100) supplemented with cOmplete proteinase inhibitor (Roche). The suspension was filtered through 40 µm nylon cell strainer

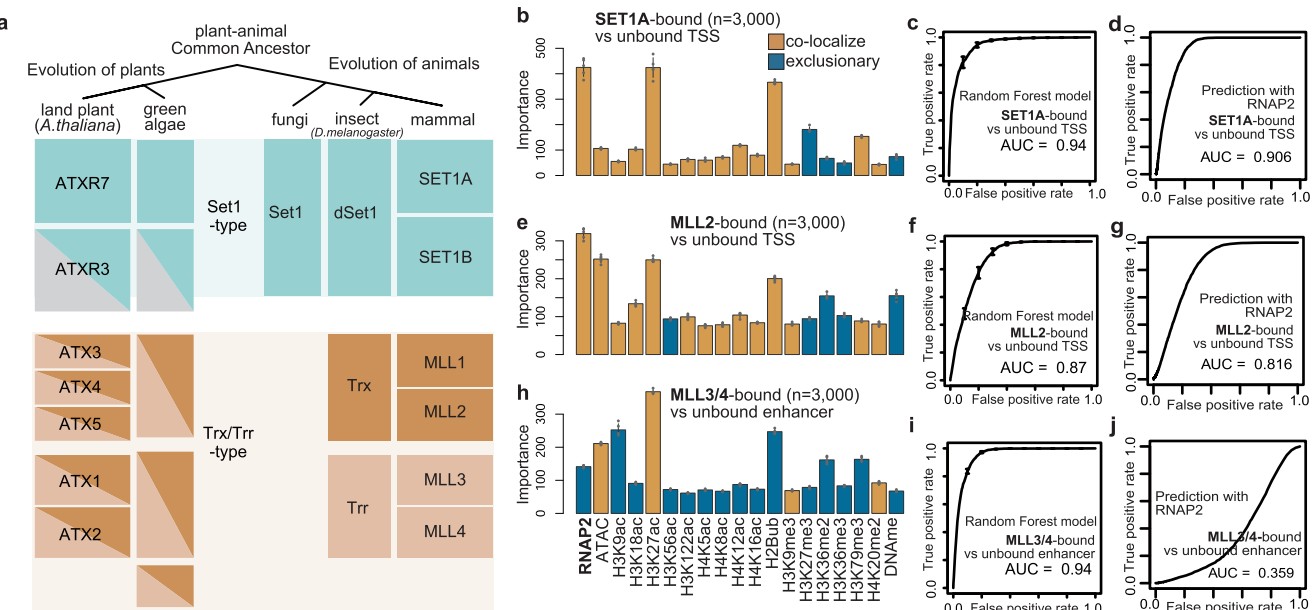

**Fig. 7 | Localization and functional mode of mammalian H3K4 methyl-transferases. a** Evolution of H3K4 methyltransferases in eukaryotes based on[18,43]. **b**, **e**, **h** Chromatin features predictive of the localizations of SET1A (**b**) and MLL2 (**e**) at the TSS regions, and MLL3/4 (**h**) at the enhancer regions. Bars are colored brown if the mean abundance of the feature is bound region > unbound region (i.e. colocalize), and colored blue otherwise (i.e. exclusive). Error bars are the standard deviation of the 5 repeats of training. **c**, **f**, **i** ROC plot showing the prediction accuracy of the random forest models corresponding to (**b**, **e**, **h**). **d**, **g**, **j** ROC plot showing the prediction accuracy using levels of RNAP2 as the sole predictor. All ROC and AUC are calculated with test data (25% of the original data). Error bars represent the standard deviation of the 5 repeats of training. All the genomic data used here are curated from previous works (Supplementary Data 7).

(Falcon) and pelleted by centrifugation at 3000 rpm at 4 °C for 5 min. The pellet was washed twice with NIB, re-suspended in Laemmli loading buffer, and boiled for 5 min. Proteins were resolved on SDS-PAGE, transferred to PVDF membrane (BioRad), and blocked with Blocking one solution (Nacalai). The loaded sample quantities were normalized with H3 or H4. When normalizing with H3, identically loaded blots were respectively incubated with H3 modification specific antibodies (H3K4me1 (ab8895; Abcam), H3K4me2 (ab32356; Abcam), H3K4me3 (ab8580; Abcam)) and H3 antibody (ab1791; Abcam) in 0.5 µg/ml dilution. When normalizing with H4, the membrane was cut horizontally between H3 and H4 using protein ladder markers as a guide. Membranes containing H3 and H4 bands were respectively incubated with H3 modification specific antibodies and H4 antibody (raised and gifted by Dr. Akihisa Osakabe). Second antibody was Anti-Rabbit IgG HRP (NA934; cytiva, 1/10^5 dilution). The blots were developed with ECL prime solutions, and the signal was quantified by iBright Imager (Invitrogen). The signal of H3 modification was divided by H4 or H3 signal to normalize for the amount of protein loaded. The WT to mutant ratios of the normalized modification signals are shown in Supplementary Fig. 2a.

**ChIP-seq**
ChIP targeting histone modifications, RNAP2, and epitope-tagged proteins were essentially carried out as described previously[14]. Briefly, approximately 1.5 g frozen seedlings were ground into fine powder and suspended in 25 ml of nuclei isolation buffer (for histone modifications, 10 mM HEPES pH7.6, 1 M sucrose, 5 mM KCl, 5 mM MgCl2, 5 mM EDTA, 1% formaldehyde, 0.1% β-mercaptoethanol, 0.6% Triton X-100, supplemented with 1 tablet/50 ml cOmplete proteinase inhibitor and 1 mM Pefabloc SC (Roche)). The buffer for epitope-tagged proteins additionally contains 0.1 mM spermine, 0.5 mM spermidine and 1.5 mM ethylene glycol bis(succinimidyl succcinate) (EGS; ThermoFisher). Also, 0.6% Triton X-100 was replaced with 0.3% IGEPAL CA-630. The suspension was incubated 10 min in RT to crosslink, quenched with 130 mM glycine, filtered through 40 µm nylon cell strainer and pelleted by centrifugation at 3000 rpm at 4 °C for 10 min. The pellet was

resuspended in 300 µl of nuclei isolation buffer and layered on top of 500 µl of nuclei separation buffer (10 mM HEPES pH 7.6, 1 M sucrose, 5 mM KCl, 5 mM MgCl2, 5 mM EDTA pH 8.0, 15% Percoll) and pelleted by centrifugation at 3000 × g for 5 min at 4 °C. For histone modifications, the nuclear pellet was lysed with 150 µl of lysis buffer (50 mM Tris–HCl, pH 7.8, 10 mM EDTA, 1% SDS) then diluted with 800 µl of dilution buffer (50 mM Tris–HCl, pH 7.8, 0.167 M NaCl, 1.1% Triton X-100, 0.11% sodium deoxycholate). For epitope-tagged proteins, the pellet was directly resuspended in 950 µl RIPA buffer (50 mM Tris-HCl pH7.8, 150 mM NaCl, 1 mM EDTA, 0.1% SDS, 1% Triton X-100, 0.1% Sodium deoxycholate and cOmplete proteinase inhibitor). The suspension was sonicated using a Covaris S220 Focused-ultrasonicator (Covaris) and milliTUBE 1 ml AFA Fiber (Covaris) with the following settings: time, 20 min; duty factor, 5%; Cycles per Burst, 200; temperature (water bath), 4–6 °C; and Peak Incident Power, 140 for histones and repeat alternately 105 and 140 for epitope tagged proteins. The sonicated chromatin was then centrifuged at 13,000 g for 3 min, and the supernatant was diluted with RIPA buffer and aliquoted. The chromatin solution was incubated with 1 µg of antibodies overnight at 4 °C. Antibodies are H3, H3K4me1/2/3 (see the section of Western blot), H3K4me0 (MABI0301, MBL, sold as anti-H3 but specific to unmodified H3K4 as shown in ref. 70), H3K36me2 (MABI0332; MBL), H3K36me3 (MABI0333; MBL), H3K27me3 (MABI0323; MBL), H2Bub (MM-0029; Medimabs), H4K16ac (07-329; Millipore), RNAP2 total CTD (MABI0601; MBL), RNAP2 phospho S2 (MABI0602; MBL), RNAP2 phospho S5 (MABI0603; MBL), FLAG (F1804; SIGMA). Then the antibody-chromatin mix was incubated for 2 h at 4 °C with magnetic beads; Dynabeads M280 Sheep anti-mouse IgG in the cases of MBL antibodies, and with Dynabeads Protein G in the cases for the others. For Histone modifications, the incubated beads were washed once with 1 ml of RIPA buffer, twice with 1 ml of High-salt RIPA buffer (RIPA buffer with 500 mM NaCl), once with 1 ml of LiCl buffer (10 mM Tris–HCl, pH 7.8, 1 mM EDTA, 0.25 M LiCl, 1% IGEPAL CA-630, 1% sodium deoxycholate) and once with 1 ml of TE buffer (10 mM Tris–HCl, pH 7.8, 1 mM EDTA), each time rotating for 10 min in 4 °C. For epitope-tagged proteins, beads were washed once with RIPA buffer, medium salt RIPA buffer

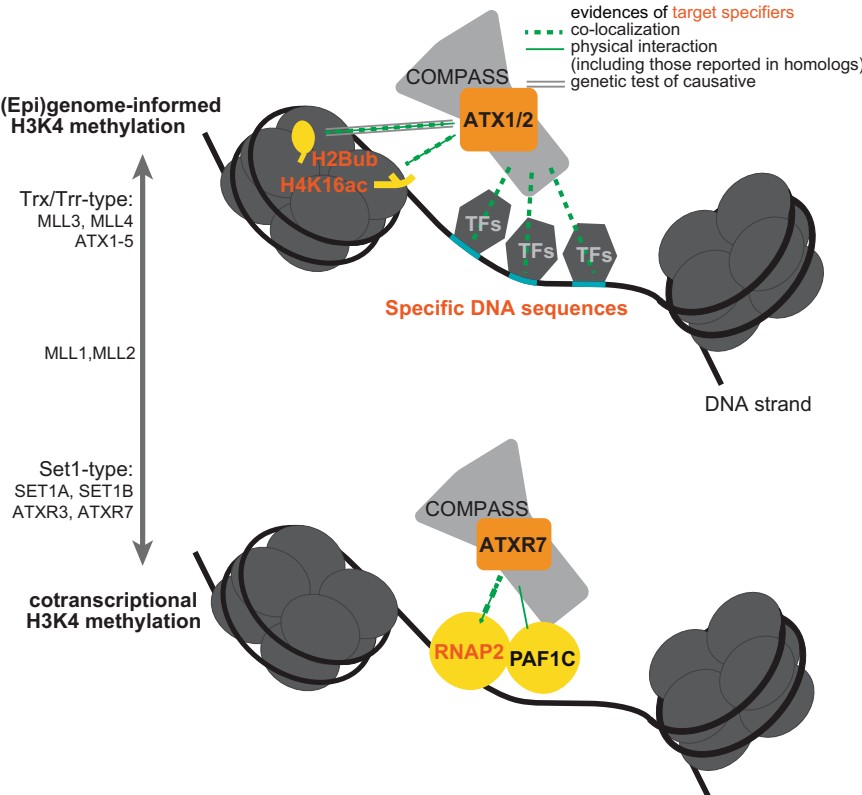

**Fig. 8 | Suggested models of (epi)genome-informed and cotranscriptional modes of H3K4 methylation.** Random forest and SVM modelings suggest that ATX1 and ATX2 localize based on chromatin features such as H2Bub, H4K16ac (Fig. 3) and several DNA sequences (Fig. 4). Mechanistic studies on ATX homologs indicate that H2Bub[53] and H4K16ac[52] tether H3K4 methyltransferases to nucleosomes. Some TFs may also bridge ATX to specific DNA sequences. On the other hand, ATXR7 is suggested to be cotranscriptional, based on its colocalization with RNAP2 (Fig. 3) and its contribution to the transcription-H3K4me1 correlation (Fig. 5). Studies in yeast homolog SET1 clarified Polymerase II Associated Factor 1 Complex (PAF1C) interacts with RNAP2 and COMPASS, thus providing a scaffold for H3K4 methyltransferases to work with RNAP2[6,7]. These (epi)genome-informed and cotranscriptional modes of H3K4 methylation may also be generalized to other Arabidopsis H3K4 methyltransferases (Fig. 6) and other organisms (Fig. 7).

(RIPA buffer with 300 mM NaCl), high salt RIPA buffer and TE buffer. DNA was eluted from the beads and reverse-crosslinked by adding 100 μl of elution buffer (10 mM Tris−HCl, pH 7.8, 0.3 M NaCl, 5 mM EDTA, 0.5% SDS) and incubating overnight at 65 °C. The DNA samples were then treated with RNase A (Nippon Gene) for 30 min and with Proteinase K (ThermoFisher) for 2 h at 37 °C, and purified using the Monarch PCR & DNA Cleanup Kit (NEB). For spike-in ChIP-seq, chromatin samples of *Arabidopsis*, adjusted to have the same concentration of DNA, were mixed with chromatin of *Schizosaccharomyces pombe* (spike-in). Chromatin of *S. pombe* was prepared as follows; cells were fixed with 1% formaldehyde for 10 min in 37 °C, quenched with glycine, resuspended in Buffer 1 (50 mM HEPES-KOH pH 7.5, 140 mM NaCl, 1 mM EDTA, 1% Triton X-100 and 0.1% sodium deoxycholate supplemented with proteinase inhibitors) and disrupted with bead shocker. The suspension was sonicated by Covaris S220 with a setting similar to that of Arabidopsis histone, except the sonication time being 15 min. The sonicated suspension was centrifuged for 5 min at 13,000 rpm at 4 °C. ChIP-seq libraries were made with KAPA Hyper Prep Kit (Kapa Biosystems), and dual size-selected using AMPureXP (Beckman Coulter) to enrich 200−500 bp fragments. The libraries were 50-bp single-end sequenced by HiSeq4000 sequencer (Illumina) in Vincent J. Coates Genomics Sequencing Laboratory at UC Berkeley, or 150 bp paired-end sequenced by the HiSeqX Ten sequencer (illumina). Biological replicates were conducted on independently grown plants.

## ATAC-seq
For ATAC-seq, nuclei preparation, tagmentation and library prep procedure followed sucrose sedimentation protocol of ref. 71.

Briefly, 100 mg of frozen seedlings were ground into fine powder, thoroughly resuspended in 10 ml of NPB (20 mM MOPS pH7, 40 mM NaCl, 90 mM KCl, 2 mM EDTA, 0.5 mM EGTA, 0.5 mM spermidine, 0.2 mM spermine, cOmplete protease inhibitor), filtered through 40 μl nylon mesh, then centrifuged for 10 min at 1200 g at 4 °C. The pellet was resuspended in 1 ml of NEB2 (0.25 M Sucrose, 10 mM Tris-HCl pH 8, 10 mM MgCl 2, 1% Triton X-100, cOmplete Protease Inhibitor) and centrifuged for 10 min at 12,000 g at 4 °C. The pellet was resuspended in 300 μl NEB3 (1.7 M Sucrose, 10 mM Tris-HCl pH 8, 2 mM MgCl2, 0.15% Triton X-100, cOmplete proteinase inhibitor), layered on top of 300 μl of NEB3, then centrifuged for 10 min at 16,000 g at 4 °C. This step was repeated twice before the pellet was suspended in NPB. Tagmentation was performed on approximately 25,000 nuclei for 30 min at 37 °C using Tagment DNA TDE1 Enzyme and Buffer Kits (Illumina), with the reaction volume of 25 μl. DNA was immediately purified, amplified with ATAC Primers[71] and NEB Next High Fidelity PCR Mix for 7 cycles, and purified with Agencourt AMPure XP. The libraries were 50-bp single-end sequenced.

## BS-seq
Whole-genome bisulfite sequencing (BS-seq) was conducted as described before[72]. DNA was extracted with Nucleon PhytoPure (cytiva) and subjected to bisulphite conversion using MethylCode Bisulfite Conversion Kit (Life Technologies). Bisulphite-treated DNA molecules were PCR amplified with 10 cycles using KAPA HiFi HotStart Uracil+ ReadyMix (Kapa Biosystems) and purified with Agencourt AMPure XP (Beckman Coulter). The libraries were 150 bp paired-end sequenced by the HiSeqX Ten sequencer (Ilumina).

## RNA-seq

Total RNA was extracted from seedling with RNeasy Plant Mini Kit (Qiagen), then polyA selected, fragmented, and made into a library with KAPA Stranded mRNA-seq Kit (Kapa Biosystems) following the manufacturer's protocol. The libraries were 50-bp single-end sequenced as described above. One sample corresponds to one individual seedling, and three samples were sequenced for one genotype.

## Genomes and annotations

For *Arabidopsis*, TAIR10 reference genome and Araport 11 gene annotations were used. In this paper, 'gene' otherwise stated refers to all the transcribed features in Araport11 annotation, including TE genes or non-coding RNAs. 'protein_coding' genes refer to the nuclear genes that are annotated as such in Araport11 annotation, excluding the ones that lack primary transcript annotation. For *Mus musculus*, we used reference genome mm9, which is not the latest, for compatibility reasons. Gene annotations are from NCBI RefSeq. Enhancer annotations are from Enhancer Atlas V2[73]. Corresponding to the cell types of the data origin, MLL3/4 data was analyzed with R1 enhancer annotation, while MLL2 and SET1A data were analyzed with V6.5. For *S. pombe*, the Ensembl_EF2 genome was used.

## Sources of reanalyzed data

For *Arabidopsis*, the genomic data were reanalyzed from the following works; H3K9me2, DRR023251[13]; H3K9ac, GSM701925 and H3K18ac, GSM701927[74]; H3.1, GSM856055 and H3.3, GSM856054[75]; H1, GSM2544793 and H2A.Z, GSM2544791[76]; H3K14ac, GSM2051285[77]; Like-Heterochromatin Protein 1, GSM2028108[78]; H3K56ac, GSM2027818[79]; H3K23ac, GSM1701017 and H4K16ac, GSM1701018[80]; H3K9me1, GSM1242411[81]; and H2B ubiquitination, GSM3092016[82]. Growth conditions and the tissue-of-origin of these data are summarized as Supplementary Data 4.

For *M. musculus*, the genomic data were reanalyzed from the following works; SET1A[44], MLL2 and its control[45], MLL3/4, its control, RNAP2, ATAC and H3K27ac[46], H4K16ac[83], H3K36me2[84], H3K9ac, H3K56ac, H3K36me3 and H3K79me3[85], H4K12ac, H3K9me3 and H4K20me2[86], H3K18ac, H3K122ac, H4K5ac, H4K8ac and H3K27me3[87], and DNAme[88]. The exact names of the analyzed files are provided in Supplementary Data 7. All data is from mESC.

## Mapping and counting coverages

For *Arabidopsis*, ChIP-seq and ATAC-seq data were processed essentially as described in[14]. Briefly, our sequenced data were mapped to the TAIR10 reference genome using bowtie[89] -v 2 -m 1 option. In cases of spike-in ChIP-seq, the sequenced data were also mapped to the *S. pombe* genome. Mapped reads for single-end sequence reads were extended to 250 bp to represent sequenced fragments, then counted for coverage over the region of interest specified as bed files, using SAMTools[90] and BEDTools[91]. The coverage was RPM/RPKM normalized with custom scripts. Metaplot profiles and heat maps were generated with ngs.plot[92]. mRNA-seq data were also processed as described in[14] using STAR[93]. BS-seq data were processed as described in[13] using Bismark[94]. For the reanalyses of the published data, SRA data were adapter trimmed following the original papers' procedure, mapped, and counted essentially the same as above.

For *M. musculus*, SRR data of ChIP-seq targeting H3K4 methyltransferases were adapter-trimmed and mapped to mm9 reference genome with -v 2 -m 1 option, then counted and normalized as described above. The other genomics data of *M. musculus*, namely the data of chromatin modifications used for the random forest, were curated and parsed from bigWig format provided as supplementary files at Gene Expression Omnibus (GEO) (Supplementary Data 7). bigWig files were first converted to bedGraph with bigWigtoBedGraph[95] of UCSC tools, then to bed file with the awk command of Unix, and when

applicable, genome version was converted into mm9 with liftOver of UCSC tools, then coverage were counted over the regions of interest.

## Clustering of the 6 single mutants

We selected the top 3000 genes that had the highest reduction of H3K4me1 (RPKM) compared to WT for each mutant. The regions from TSS to TTS of the 3000 genes were split into 6 bins. For the WT and the six mutants, the number of reads mapped to each of the 6 sub-gene regions were counted, RPM normalized, and the difference between WT and the mutant was calculated. To capture the trend of which ones of the 6 bins lose H3K4me1, 6 bins in each one gene were standardized. Thus, for each mutant, a matrix of 6 regions x 3000 genes was obtained. These matrices were vectorized, and Euclid distances were calculated, which were used for hierarchical clustering (Fig. 1b). To explain the idea behind this clustering, considering if a hypothetical methyltransferase catalyzing H3K4me1 into H3K4me2 is lost, H3K4me1 level would be elevated in the region that H3K4me2 originally occupied. We reasoned that similarity in affected sub-gene regions reflects the similarity in function.

## Peak calling

For *Arabidopsis*, differentially modified regions were identified by comparing mutant to WT using MACS2[96] with option -g 1.3e8. Peaks detected in centromeres are removed. Peaks were assigned to overlapping genes by bedtools 'intersect' with default options. If a peak is assigned to multiple genes, all genes are kept. If multiple peaks are assigned to one gene, one gene is added to the list.

Enrichment peaks of H3K4 methyltransferases were identified by comparing against negative control (ChIP-seq with FLAG antibody in non-transgenic WT) using MACS2[96] with two settings; the default (-g 1.3e8) and more relaxed option (-g 1.3e8 -q 0.3 –nomodel). Both counts are provided as Supplementary Data. 3. The latter relaxed peaks are used to measure the distances of the peak summits of from the nearest TSS (in the cases of ATX1, ATX2, and ATXR3) or TTS (in the case of ATXR7), which were visualized as a histogram (e.g. Fig. 2d). Based on the histogram, we identified the region that ATX1, ATX2 typically occupy to be 150 bp upstream and 300 bp downstream of TSS, which we named 'TSS region'. Similarly, 'TTS region' is defined as 200 bp upstream to 200 bp downstream of TTS, where ATXR7 localizes, and 'TSS region 2' starts from 200 bp downstream of TSS to further 400 bp, where ATXR3 localizes. For *M. musculus*, enrichment peaks of H3K4 methyltransferases were called with default options of MACS2. The distances of the summits of the called peaks from the nearest TSS or the midpoint of enhancer region (in the cases of MLL2 and SET1A, V6.5 enhancer, while in the case of MLL3/4, R1 enhancer) were measured and visualized as a histogram. Based on the histogram, we identified the region that SET1A and MLL2 typically occupy to be 150 bp upstream and 300 bp downstream of TSS, coinciding with the 'TSS region' of *Arabidopsis*. Similarly, we defined 'enhancer region', where MLL3/4 localizes, to be 900 bp on both sides from the center of Enhancer Atlas's annotation.

## Spike-in normalization

The summary statistic R, plotted in Supplementary Fig. 1d was calculated as follows; R = ratio of the number of reads that mapped to *Arabidopsis* and to *S/S.p.*). $R_{WT\_H3} : R_{mutant\_H3} = R_{WT\_H3K4me1}$ : Expected $R_{mutant\_H3K4me}$. (Actual $R_{mutant\_H3K4me1}$)/(Expected $R_{mutant\_H3K4me1}$) for each mutant was plotted in the bar graph. The raw counts of reads are summarized in Supplementary Data 2.

## Random forest

For ATX1, ATX2, and ATXR7 models, genes shorter than 500 bp were filtered out. Gene was divided into three regions; TSS region, TTS region (definitions described above), and "gene body" region. The "gene body" region is the transcribed region minus TSS and TTS

regions. Similarly, for ATXR3 models, the transcribed regions of genes longer than 800 bp were divided into TSS, TSS2, TTS, and "gene body2" regions. "Gene body2" region is the coding region minus TSS, TSS2, and TTS regions.

Explanatory features are levels of chromatin modifications/chromatin states calculated over these three regions per gene. We generated data from samples under the same conditions as ATX(R) ChIP for some modifications that seemed to be particularly variable or well-studied (Expression, RNAP2, H3K27me3, H3K36me3, DNAm) to minimize tissue/stage derived variation. For the other modifications, raw read files reported by preceding works (see 'sources of the reanalyzed data' section and Supplementary Data 4) were used. For "gene body" and "gene body2" regions, both length-normalized and unnormalized values are included. These data combined with expression level (FPKM of mRNA-seq in WT, from this work) and gene length were used as the predictor variables.

Objective variables were set as two classes; ATX(R)-bound and -unbound genes. ATX(R)-bound genes are defined as the top 3000 genes that have the highest ChIP-seq signal (RPM values of ATX(R)-Tag expressing lines - negative control) in the typical occupying region defined from the above-mentioned peak calling; TSS region for ATX1 and ATX2, TTS region for ATXR7, TSS2 region for ATXR3. ATX(R)-unbound genes are the rest of protein coding genes ($n$ = all protein coding genes longer than 500 or 800 bp − 3000). We uniformly set the number of bound genes to 3000, at the potential expense of predictive accuracy, in order to reduce parameter variance between models.

To balance the two classes, 3000 genes out of the ATX(R)-unbound genes were randomly selected, and the data on chromosome 1 to 4 were used for training random forest (R package 'RandomForest'), while data on chromosome 5 were held out as test data.

For SET1A, MLL2, and MLL3/4 models, explanatory features are levels of chromatin states calculated over TSS regions or enhancer regions, using previously reported data in mESC (see 'sources of the reanalyzed data' section). Objective variables were set in the same manner as ATX(R)s models; that is, Set1/MLL2/MLL3/4-bound regions were defined as the top 3000 regions with the highest TSS region signals regions or enhancer regions, and unbound regions were downsampled into 3000. 25% of the data were held out from the training and used as test data.

Both for *Arabidopsis* and *M. musculus*, the random sampling and training steps were repeated 5 times, and the average and the standard deviation of the importance derived from the 5 models were plotted. ROC was plotted based on the prediction of the 5 models, using test data.

In order to simulate ATX1/2 localization in the absence of H2Bub, a feature matrix, in which all H2Bub data was replaced with 0, was given to the trained random forest models to make predictions. 'Genes that lose ATX1 in the in silico hub' was defined as genes that are predicted to be ATX1-bound at least once among the 5 models that constitute the original random forest model, and predicted to not to be bound at least once among the 5 models that are given the H2Bub-0 feature matrix.

## Support vector machine

Application of SVM to DNA sequence essentially followed previous work on human enhancer classification[97]. Briefly, DNA sequences on the coding strand of TSS regions were converted to k-mer frequency vectors of 4^k features. The frequency vectors were standardized and given bound or unbound labels in the same way as random forest.

The data on chromosome 5 were held out as test data. Data on the chromosome 1 to 4 were used for 5 cross-fold validation for training to select the optimal parameters. For each trial of cross-fold validation, data of the unbound class were randomly sampled independently to balance the two classes. The averaged weights of features (k-mer sequences), which were calculated based on the 5 independently trained models using the best parameters, were used for further motif annotations. ROC was plotted based on the prediction of the 5 models with test data. We trained 5,6,7-mer lSVM or 6-mer kernel-SVM to find that the accuracy of 6-mer lSVM was about the same or better than the rest (Supplementary Fig. 18).

## Motif annotation

The decision function of linear SVM is the sum of weighted features (in this case, standardized occurrence of each k-mers), which allows us to interpret the highly weighted motifs to be predictively important ones. Using the weights averaged from the 5 best models, sixty 6-mers that had the largest positive weights, largest negative weights, and smallest absolute weights, and sixty random 6-mers were selected.

To make clusters of related motifs within each group of sixty 6-mers, similarity scores were calculated between all pairs of the 6-mers. Similarity scores are defined as; reverse complement (=1) > 1 nt offset > 1 nt offset on the opposite strand > the others (=0). The resulting similarity score matrices were visualized with the qgraph[98] package in R.

Next, to annotate the 6-mers while considering the context the 6-mer is located, we calculated the ATGC frequency in the flanking three bases on both sides of each 6-mer. The flanking ATGC frequency of the positively weighted sixty 6-mers was calculated with the sequence of ATX-bound TSS region, while negatively weighted and random/near-zero weighted 6-mers were calculated with the sequence of ATX-unbound and random TSS region, respectively. The resulting probability matrices representing 3 nt + 6 nt + 3 nt were converted into meme format and searched for matching motifs using TOMTOM against all the *Arabidopsis* databases included in the MEME Suite's motif_databases.12.19[99]. For each motif, TOMTOM hits were summarized in Supplementary Data 6, and top-hit satisfying q.value< 0.1 is indicated by colors representing TF families (e.g. Fig. 4e, f). The distribution of the motifs of interest, i.e., GAGA-stretch, TATA-box, telo-box, and RGCCCAW, was visualized by the presence of the 6-mers comprising the motifs. Namely, {AAGAGA, GAGAGA, AAGAGG, GAGAGG, AGAGAA, GGAGAA, AGAGAG, GGAGAG} represented GAGA-stretch, {TATAAA, TATATA, ATATAA, ATAAAT, TAAATA, ATATAT, TTATAA, TTATAT} represented TATA, {AAACCC, AACCCT, ACCCAA, ACCCTA} represented telobox and {AAGCCC, AGGCCC, GGCCCA, AGCCCA, GCCCAA, GCCCAT, GGGCTT, GGGCCT, TGGGCC, TGGGCT, TTGGGC, ATGGGC, AATGGG, CCCATT} represented ARGCCCAWT.

The PhastCon[100] score calculated by ref. [101], based on 63 angiosperm species was averaged per nucleotide on each 6-mer appearing in the directional sequence of the TSS region of all *Arabidopsis* genes as an index of sequence conservation.

## Correlation between H3K4me and transcription

Spearman correlation between mRNA-seq and ChIP-seq data of H3K4me was calculated using all protein-coding genes. mRNA-seq data was FPKM normalized, ChIP-seq data targeting H3K4me1 was RPKM, H3K4me2/H3K4me3 was RPM normalized. For color-filled contour plots, genes for which no transcription was detected were excluded.

## Reporting summary

Further information on research design is available in the Nature Research Reporting Summary linked to this article.

# Data availability

The high-throughput sequencing data generated in this study is available in the NCBI database under the accession number PRJNA732996. Source data are provided with this paper.

## Code availability

Codes are available on GitHub (https://github.com/Satoyo08/Arabidopsis_H3K4me1).

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

## Acknowledgements

We thank all Kakutani laboratory members for helpful discussion and technical assistance. This work used the Vincent J. Coates Genomics Sequencing Laboratory at UC Berkeley, supported by NIH S10 OD018174 Instrumentation Grant. The computations were partially performed on the NIG supercomputer at NIG, Japan. We thank NASC/ABRC for distributing the seeds. The *ashr3* mutant was kindly gifted by Taku Sasaki. This work was supported by grants from Japan Science and Technology Agency (JST) PRESTO (no. JPMJPR17Q1) to S.I., JST CREST (no. JPMJCR15O1) to T.K., and Japan Society for the Promotion of Science (JSPS) (nos. JP26221105, JP15H05963 and JP19H00995 to T.K., JP20H05913 and JP22H02299 to S.I. and JP19J21882 to S.O.). S.O. is supported by the JSPS Research Fellowship for Young Scientists.

## Author contributions

S.O., T.K. and S.I. conceived the study. S.O., M.T., K.T., and S.I. performed the experiments. S.O. conducted the analysis. S.O. drafted and S.O., T.K. and S.I. edited the manuscript.

## Competing interests

The authors declare no competing interests.
