## [Peer Review File · Nature Communications]

Transcription-coupled and epigenome-encoded mechanisms direct H3K4 methylationREVIEWER COMMENTS

Reviewer #1 (Remarks to the Author):

Oya et al presented an interesting study on the functions of ATX family in H3K4methylation, with a focus on the role of ATX1, ATX2, and ATXR7 in H3K4me1 marking of transcribed gene spaces, by creating single and higher order mutants for the ATX families and characterizing the epigenome of these mutants, as well as binding profile of select ATXs. The integrated analyses, highlighted with machine learning approaches, uncovered labor division and different localization rules within the ATX family. While highly interesting and innovative, the study is weakened by the lack of rigorous statistics and experimental support of cause-and-effect statements drawn on machine learning analysis, as detailed below:

1) Some basic claims are not approached in a statistically rigorous manner. Some conclusions are based solely on visual speculation of figures. A few examples are listed below:

The H3K4methylation epigenome of mutants: There seems to be only one replicate of ChIP-seq, therefore, it is hard to differentiate biologically meaningful difference from experimental noise. Also, it is not clear whether proper background control (input DNA or H3 pulldown) has been used for calling methylation islands in ChIP-seq analysis. The comparison of epigenomes between mutants and WT seem to rely too much on eyeballing. For example, the statement on Line 76-77: "atx 2 and atxr 7 showed large decrease in H3K4me1 level (extended Fig. 1a)" would be better supported by determining how many genes passed statistical cutoff as being differentially methylated between WT and mutant in each of the comparison. I have similar concerns about the claims made based on eyeballing in extended figure 1c.

The statement that H3K4me is mostly decreased in the triple mutant (Line 88-91) is only supported by the western blot in Figure 1e, which is not a quantitative method, especially given there is whole genome profiling data in this study. The whole genome profiling in figure 1c and 1d suggest that there are a substantial amount of genes with increased H3K4me1 in the triple mutant compared to WT. Therefore, a more in-depth analysis is needed to clarify (e.g. how many genes with increased H3K4me vs how many genes with decreased H3K4me, the fold change of each, etc)

The ATX1, 2, R7 binding ChIP-seq seem to be replicated but the replicates are not synthesized in data analysis. It is unclear how many genes are bound by each ATX protein. The statement that "genes showing H3K4me1 loss in atx1/2/r7 mutants tended to be bound by ATX1, 2, and R7" on line 109 is only supported by eyeballing supplemental Figure 2, which does not serve as a strong support because there are also many ATX1/2/7 bound genes with increased H3K4me1 in this figure. A better test would be to calculate the significance of the overlap between the hypomethylated/hypermethylated genes in the triple mutant with the bound genes, which the author did not provide.

A few statements on correlation are only supported by scatterplots without actually calculating the correlation, examples are Extended Figure 3a and Figure 4d. In Figure 5, the correlation is quantified, but with the overall correlation being low, it is an overstatement to say that there is a big difference between the 0.23 in WT and 0.22 in atxr7 mutant, thus the subheading of this part of result "the positive correlation between transcription and H3K4me1 is mediated by ATXR7" is not well supported.

The statement "ASHH3 functions downstream of ATX1/2/R7 marked H3K4me1 to mediate H3K36me3" in line 135-137 is not supported by Extended Figure 3c. A comparison of the genes that are hypomethylated with H3K36me3 in ashh3 with the ATX1/2/R7 target genes are a more direct evidence.

2) While the performance of the machine learning is impressive, I have a few concerns about the machine learning approach and its interpretation.

First, since the chromatin and genomic features (Pol II, other histone marks) used for machine learning are from a different source, the author should comment on whether they are collected from similar tissue and developmental stages, since these could be quite tissue and developmental/environmental dependent.

Second, I am wondering if the choice of an arbitrary and static number of "marked" genes instead of statistically rigorously defined bound genes affects the result.

Third, the machine learning results suggest colocalization but not the mechanism of recruitment. Therefore, the directionality or cause-and-effect seem to be overstated, if not backed up by further biochemical or genetic evidence. For ATXR7 and ATXR3, the author suggests that they are co-transcriptionally recruited to deposit H3K4 methylation because their binding to genes could be best predicted by the localization of Pol II. However, it is also possible that ATXR3/7 function first to deposit H3K4 methylation, which directs the binding of Pol II. This alternative hypothesis could also lead to the observation in Figure 6e-f. Further genetic or biochemical analysis is needed to support that ATXR3 and ATXR7 are recruited co-transcriptionally. Same for the claim that ATX1/2 are likely recruited by chromatin features and cis-motifs.

Finally, in the SVM motif analysis, since ATXR7 binds to the TTS, it is not surprising that TSS motifs are not good predictors. The author should also consider testing with TTS 6-mers.

Minor issues:

Introduction Line 37-41: needs elaboration and clarification.

Line 54 "methylase mutants with decreased H3K4me1 levels have not been reported" and line 58 "by increasing H3K4me1/2/3 level within the FLC locus" are conflicting. Please clarify.

Supplemental figure 2: the plots at the bottom need better marking and explanation in legend.

It might be more conventional and easier to read to change "methylase" to "methyltransferase"

The clustering in Fig. 1B has no scale.

Extended Fig 1b pattern agrees with the clustering in Fig 1b very well and supports the logic of making atx1/2 double mutant. Overall I feel extended Fig 1b is a more informative and powerful figure to include than figure 1b.

Figure 1f: I don't understand why there is only one expected number of intersections, not three for the three possible pairwise comparisons?

Fig 4e and 4f are a little hard to read, highlighting different circles for GAGA, TATA and telobox would be helpful.

Figure 8 has no legend and the idea that ATXR7 is recruited through COMPASS/PAF1c should be explained.

Reviewer #2 (Remarks to the Author):

This study identified ATXR1/2/r7 as H3K4me1 methylases. They compared H3K4me1 ChIP data in atx signal and triple mutants, and found that atx 1/2/r7 triple mutants have dramatic decrease in H3K4me1. In addition, the authors analyzed ATXR1, 2, r7 ChIP data using random forest algorithm and found that ATXR7 is correlated with RNAP2, while ATXR1 and 2 are associated with other chromatin modifications. In general, this study demonstrated a role of ATXR1/2/R7 in accumulating H3K4me1. However, some data seems not consistent with each other, and several conclusion were not well supported by the data. I have several concerns:

1. Results shown in figure 1c and figure 1e are not consistent. In figure 1c atx1/2/r7 had much stronger signal than atx1/2, however, in figure 1e, the H3K4me1 accumulation signals are similar in atx1/2/r7 and atx1/2. How do you explain this difference?
2. Figure 2b, the researchers showed that ATX1 and ATX2 are preferentially located in the 5' end of gene, while ATX7 prefers to 3' end of gene. I suggest them to examine whether the H3K4me1 is decreased at the 3'end in atx1/2 double mutant, but is decreased at the 5'end in atxr7 single mutant.
3. An important and unanswered question about H3K4me1 is whether it is the reason or the result of transcription. In figure 5A and B, the author plotted H3K4me1 signal vs gene expression in WT, atx1/2, and atxr7. They concluded that the positive correlation between transcription and H3K4me1 was disturbed by atxr7. However, a more interesting analysis could be that whether the genome-wide loss of H3K4me1 are associated with the change of gene expression in atxr1/2, atxr7 or atxr1/2/7. For example, if there is a gene that is actively expressed and is enriched with H3K4me1 in WT, when the H3K4me1 in this gene is lost in atx1/2/r7 mutants, is this gene expression lost or unchanged in atx1/2/r7 mutants?

Minor concerns:

- 1: line 85, figure 1c, the author concluded that atx1/atx2 double mutant had stronger effect than that in atx1 and atx2 single mutants. However, I think in figure 1c, the signals in atx1 and atx2 single mutants and atx1/atx2 double mutant are quite similar.

Reviewer #1 (Remarks to the Author):

Oya et al presented an interesting study on the functions of ATX family in H3K4methylation, with a focus on the role of ATX1, ATX2, and ATXR7 in H3K4me1 marking of transcribed gene spaces, by creating single and higher order mutants for the ATX families and characterizing the epigenome of these mutants, as well as binding profile of select ATXs. The integrated analyses, highlighted with machine learning approaches, uncovered labor division and different localization rules within the ATX family. While highly interesting and innovative, the study is weakened by the lack of rigor statistics and experimental support of cause-and-effect statements drawn on machine learning analysis, as detailed below:

We thank the Reviewer #1 for the positive evaluation of our manuscript and constructive suggestions. We think that the manuscript greatly improved by incorporating the suggestions including statistical tests and experimental support of our proposal as described below.

1) Some basic claims are not approached in a statistically rigorous manner. Some conclusions are based solely on visual speculation of figures. A few examples are listed below:

The H3K4 methylation epigenome of mutants: There seems to be only one replicate of ChIP-seq, therefore, it is hard to differentiate biologically meaningful difference from experimental noise.

In response to this comment, this revision includes two sets of ChIP-seq replicates of H3K4 methylation mutants, namely,
series 1: { WT, *atx1*, *atx2*, *atxr7*, *atx12*, *atx1r7*, *atx2r7*, *atx12r7*} x {H3K4me1, H3} and
series 2: {WT, *atx12r7*, *atx345*, *atxr3*} x {H3, H3K4me1, H3K4me2, H3K4me3}.
These data are analyzed in accordance with your advice, as described below. Spike-in normalization was applied for series 1 datasets.

Also, it is not clear whether proper background control (input DNA or H3 pulldown) has been used for calling methylation islands in ChIP-seq analysis.

As a background control, we have performed H3 ChIP-seq along with H3K4me ChIP-seq to find that rarely any regions are differentially occupied by H3 among genotypes in both replicate series 1 and 2, validating the consistency among chromatin samples. We added this analysis to Supplementary Table 1.

The comparison of epigenomes between mutants and WT seem to rely too much on eyeballing. For example, the statement on Line 76-77: “*atx 2* and *atxr 7* showed large decrease in H3K4me1 level (extended Fig. 1a)” would be better supported by determining how many genes passed statistical cutoff as being differentially methylated between WT and mutant in each of the comparison. I have similar concerns about the claims made based on eyeballing in extended figure 1c.

Thank you for your helpful suggestions. Heeding your advice, we quantified the differentially modified regions using a statistical model (MACS2) in two ways;

1. We counted differentially H3K4 methylated regions by comparing every mutant to WT. As expected, the numbers of the H3K4me1-reduced regions were *atx3,atx4,atx5* < *atx1, atx2, atxr7* << *atx12,atx1r7, atx2r7* << *atx12r7*.
2. As a control, we analyzed H3 occupancy, as mentioned above.

We included these counts in Supplementary Table 1. We believe that these additions improved readability and credibility.

The statement that H3K4me is mostly decreased in the triple mutant (Line 88-91) is only supported by the western blot in Figure 1e, which is not a quantitative method, especially given there is whole genome profiling data in this study. The whole genome profiling in figure 1c and 1d suggest that there are a substantial amount of genes with increased H3K4me1 in the triple mutant compared to WT. Therefore, a more in-depth analysis is needed to clarify (e.g. how many genes with increased H3K4me vs how many genes with decreased H3K4me, the fold change of each, etc)

The normalized signal obtained by ChIP-seq is the *fraction* of reads that map to a region of interest (in this case genes) out of all the reads (expressed as RPM or RPKM). Therefore when a large decrease occurs in a part of the genome, the signal in the other parts will be increased, and vice versa. We reasoned that the decreases are the primary effects from the following reasons;

1. The total amount of H3K4me1 decreases according to western blot (Fig.1e)
2. In the *atx1, atx2, and atxr7* single mutants, as well as in the *atx1/atx2, atx1/atxr7* and *atx2/atxr7* double mutants (Extended Figure. 1a,b), H3K4me1 is reduced in a small number of genes (the smallness of the change explains the lack of false increase). Multiplexing these H3K4me1-losing mutations will likely result in changes in the same direction (decrease).
3. H3K4me0 ChIP-seq of *atx12r7* shows that where H3K4me1 signal decreases, H3K4me0 increases, consistent with the idea that ATX12R7 mediate monomethylation of H3K4. This result was added to the Supplementary Figure.1b.
4. To further validate the point, we additionally measured H3K4me1 by performing ChIP-seq replicates series 1 with spike-in normalization. We detected a decrease of the % arabidopsis reads roughly in the order of *atx1,2 atxr7* single mutants < double mutants < triple mutant, with the exception of *atx1/r7* which showed less decrease than *atxr7*, perhaps reflecting experimental noise. (Extended Figure.1c).

In summary, although we cannot prove that no genes gain H3K4me1 in *atx12r7*, we reasoned that the above results strongly suggest that the decreases of H3K4me1 are primary and the apparent increases in the other genes are due to the total read normalization.

The ATX1, 2, R7 binding ChIP-seq seem to be replicated but the replicates are not synthesized in data analysis.

Instead of integrating the two replicate datasets, we trained models for each replicate and showed consistent key conclusions. We reasoned that these independent tests provide a more rigorous examination of the robustness of the conclusion than, for example, single

training with highly confident, super-clean gene lists that were identified to be bound in both replicates. In any case, we have trained random forests with such synthesized highly confident gene lists and validated that the results are similar. (please see the rightmost column of the figure below. The other columns will be further discussed in response to your point 2-2.)

It is unclear how many genes are bound by each ATX protein. The statement that “genes showing H3K4me1 loss in *atx1/2/r7* mutants tended to be bound by ATX1, 2, and R7” on line 109 is only supported by eyeballing supplemental Figure 2, which does not serve as a strong support because there are also many ATX1/2/7 bound genes with increased H3K4me1 in this figure. A better test would be to calculate the significance of the overlap between the hypomethylated/hypermethylated genes in the triple mutant with the bound genes, which the author did not provide.

Thank you for this suggestion. We provided the number of detected ATX(R)s enrichment peaks when compared against the negative control (non-transgenic plant IPed with FLAG antibody) using MACS2 with the default setting, as Supplementary Table 3.

Of note, this peak call is different from the calls used in Fig. 2d and Supplementary Fig. 3c (also provided in Supplementary Table 3 as “relaxed setting”). For these figures, peaks were uniformly called with a relaxed option ($-q\ 0.3\ --nomodel$), because the stricter default option does not pass enough peaks to analyze peak position distribution for ATX1 ChIP-seq replicate 2. The trend that only ATX1 shows fewer peaks might reflect a low expression level of ATX1 (about one-fifth of ATX2).

The reviewer might feel concerned that the number of called peaks differs between two replicates. Because the peak number is very sensitive to data quality and sequencing depth, the peak number is merely one way to describe a particular dataset, rather than an estimation of actual targets. Although the number of the peaks fluctuates, the replicates qualitatively show good consistency (Fig. 3 and Supplementary Fig. 3 show similar trends. The peaks overlap between replicates as shown in Supplementary Fig. 2c in this revision). In addition, we calculated the overlap between hypo-H3K4me1 genes and ATX(R)-bound genes and found significant enrichment overall (Supplementary Fig. 2c). On the contrary, hyper-H3K4me1 genes are significantly underrepresented in ATX(R)-bound genes (enrichment = 0.3~0.6, $p\text{-val} < 1e-16$), further suggesting that hypo-H3K4me1 is primary, as argued above.

A few statements on correlation are only supported by scatterplots without actually calculating the correlation, examples are Extended Figure 3a and Figure 4d.

Thank you for pointing it out. The correlation estimates are now included in Figure 4d, Extended Figure 3a, and Supplementary Figure 4 cd.

In Figure 5, the correlation is quantified, but with the overall correlation being low, it is an overstatement to say that there is a big difference between the 0.23 in WT and 0.22 in *atxr7* mutant, thus the subheading of this part of result “the positive correlation between transcription and H3K4me1 is mediated by ATXR7” is not well supported.

Please let us clarify; in the previous manuscript, the correlation should be compared within panel a (*atxr7* and WT) and within b (*atx1/2* and another WT) respectively, because the heatmaps in the previous panel a and b are calculated with independent ChIP-seq data series. Therefore, the proper control for *atxr7* (0.22 ± 0.00) was 0.26 ± 0.03 . The difference is statistically significant ($p\text{-val} = 0.0084$ of Welch Two Sample t-test). Similarly, the control for *atx1/2* (0.38 ± 0.01) was 0.23 ± 0.05 , $p\text{-val} 2.5e-05$.

For this revision, we generated a series of ChIP-seq replicates as described above, which allowed us to compare WT-*atx1/2* in one go. Consistent with the previous comparisons (WT > *atx1/2* and *atx1/2* > WT), the correlation estimates were *atx12* > WT > *atx1/2* ($0.34 \pm 0.00, 0.40 \pm 0.01, 0.47 \pm 0.01$, each comparison being statistically significant). Fig.5 is now replaced with heatmaps using these new data. Former Fig.5 was moved to 'Supplementary Fig 8'. The statistical tests were included for both Fig.5 and Supplementary Fig. 8.

The concern of the Reviewer is that the overall correlation and the difference are small, but the smallness is rather expected; Firstly, the correlation itself being low is reasonable considering some of H3K4me1 is ruled by some cues other than transcription as indicated in this study. Secondly, the difference between genotypes must be slight, for H3K4me is lost on about a few thousand genes as shown in Extended Fig.1, while the correlation coefficient was calculated with basically all genes (~27,000 genes). Although the correlational differences would appear more pronounced when calculated with selected genes that lose H3K4me1, we chose all-genes-calculations in order to strictly avoid over-manipulation of data.

On a related note, it is with the same intention, to minimize room for feature engineering and to detect robust patterns that appear without any intentional 'fine-tuning' of the threshold, that we made ourselves use an arbitrary and static number (3,000 for 'bound genes' and 'marked genes') as a default in this paper.

The statement "ASHH3 functions downstream of ATX1/2/R7 marked H3K4me1 to mediate H3K36me3" in line 135-137 is not supported by Extended Figure 3c. A comparison of the genes that are hypomethylated with H3K36me3 in *ashh3* with the ATX1/2/R7 target genes are a more direct evidence.

Thank you for this advice. As you suggested, we compared ATX1/2/R7-marked genes and genes hypo-H3K36me3 in *ashh* mutants. Only in the *ashh3* mutant the enrichment was significant, supporting that ASHH3 functions downstream of ATX1/2/R7-marked H3K4me1 to mediate H3K36me3. We further validated the results with replicates of H3K36me3 ChIP-seq of *ashh* mutants ('replicate 2 were newly grown and ChIP-ed for this revision').

2) While the performance of the machine learning is impressive, I have a few concerns about the machine learning approach and its interpretation.

First, since the chromatin and genomic features (Pol II, other histone marks) used for machine learning are from a difference source, the author should comment on whether they are collected from similar tissue and developmental stages, since these could be quite tissue and developmental/environmental dependent.

To minimize tissue/stage-derived variation, we had generated data in our laboratory for features that are particularly variable (Expression, RNAP2, H3K27me3, H3K36me3, DNAm) from samples under the same conditions as ATX(R) ChIP-seq. As for the other modifications such as acetylation and histone variants, the genome-wide profiles appear only in a few papers; therefore it was difficult to estimate the magnitude of variation between tissues or to select data from similar samples. We added the table of tissue-of-origin of these

public data (Supplementary Table 4) to facilitate the readers' interpretation and added the above explanation in the Material and Methods section.

As for Random Forest analysis of *Mus musculus*, all the data are curated from mESC as noted in Material and Methods, and Supplementary Table 7.

Second, I am wondering if the choice of an arbitrary and static number of "marked" genes instead of statistically rigorously defined bound genes affects the result.

We have trained Random Forest models with multiple thresholds and confirmed the results are consistent. Either models trained with rigorous definition (genes that have significant ATX(R) peaks within TSS(TES) region in both replicates) or relaxed definition (top 6,000 genes with highest ATX(R) ChIP-seq signal in TSS(TES)) are accurate, had high importance for RNAP2 in ATXR7 models or H3K36me3, H2Bub and H4K16ac, etc in ATX1,2 models. The performance and 'importance' in each model are now reported in the above figure.

On the way of confirming above we noticed that models for ATX1/2 in Fig.3 were mixed up with another related model; in the previous manuscript, Fig.3 b,c actually show models trained to predict the absence of ATX1,2 (3,000 genes with the least amount of ATX1,2 proteins). We replaced it with the right figure. However, the corrected results are almost indistinguishable from the previous ones, and the fact that the absence of ATX is predicted by the (absence of) similar features is telling that the relationships between ATX1,2 and chromatin features are not limited to a few genes, but are genome-wide. Also, in case the readers get curious about a similar matter, we posted the R scripts for this training in GitHub so that readers can easily tweak and test on their own.

Third, the machine learning results suggest colocalization but not the mechanism of recruitment. Therefore, the directionality or cause-and-effect seem to be overstated, if not backed up by further biochemical or genetic evidence. For ATXR7 and ATXR3, the author suggest that they are co-transcriptionally recruited to deposit H3K4 methylation because their binding to genes could be best predicted by the localization of Pol II. However, it is also possible that ATXR3/7 function first to deposit H3K4 methylation, which directs the binding of Pol II. This alternative hypothesis could also lead to the observation in Figure 6e-f. Further genetic or biochemical analysis is needed to support that ATXR3 and ATXR7 are recruited co-transcriptionally. Same for the claim that ATX1/2 are likely recruited by chromatin features and cis-motifs.

Thank you for this vital comment. We share your concern and tested some of the causality and directionality. We think that the addition greatly improved the manuscript;

H2Bub directs ATX1

Firstly, we examined if H2Bub promotes the presence of ATX proteins, both by *in silico* simulation and *in planta* test (Extended Fig. 4a). We simulated ATX localization in absence of H2Bub (actually, *hub1* and *hub2* mutants in *Arabidopsis* lose H2Bub, so we call this as '*in silico hub*') by giving the already trained random forest model (models in Fig.3) a modified feature matrix, in which all H2Bub entries are replaced with 0 (Extended Fig. 4a). We detected a few hundreds of 'genes that lose ATX in *in silico hub*', which are the genes that are predicted to be ATX bound in the original model (WT model), but predicted to be unbound in the *in silico hub*.

For the *in planta* test, ATX2-tag/*atx2* was crossed with a *hub2* mutant, in which H2Bub is depleted (Liu Y 2007, leury D, 2007, Bourbousse C, 2012, Fiorucci 2019). Triple homozygotes of ATX2-tag, endogenous *atx2*, and *hub2* allele were selected and ATX2 location was examined with ChIP-seq for ATX2-tag/*atx2/hub2*, and ATX2-tag/*atx2/HUB2* (control), in two replicates.

In the *hub2* plants, we detected more genes that lost ATX2 (n=5,888 and 653 in two replicates, detected by MACS2) than gained (n=93 and 89, respectively). Those genes that lost ATX2 *in planta* significantly overlap with 'genes that lose ATX2 in *in silico hub*' (Extended Fig. 4c, p-val=8.8E-06 and 3.5E-03). The amount of *in planta* ATX2 ChIP-seq signal on 'genes that lose ATX2 in *in silico hub*' is more prominently reduced in *hub2* compared to other gene groups (Extended Fig. 4 b). These results are consistent with the view that H2Bub promotes the localization of ATX protein.

We further tested if the loss of H2Bub negatively impacts H3K4me1. To this end H3K4me1 ChIP-seq in *hub1* and *hub2* was performed with two replicates. There were more hypo-H3K4me1 genes than hyper H3K4me1 genes in *hub* (see the table below).

		number of differentially modified genes in hub	
		decrease	increase
H3 (background control)	hub1	9	1
	hub2	9	1
H3K4me1 replicate 1	hub1	1094	110
	hub2	764	165
H3K4me1 replicate 2	hub1	937	830
	hub2	1225	830

Hypo-H3K4me1 genes in *hub1/2* significantly overlap with genes that lose ATX *in planta* or *in silico* (Extended Fig. 4d), consistent with the view that H2Bub → ATX1,2 → H3K4me1.

Conversely, to test if H3K4me1 directs H2Bub we examined H2Bub deposition in *atx12r7* mutant by ChIP-seq in two replicates. Hypo-H2Bub genes significantly overlap with ATX12R7-marked genes in both replicates, suggesting H2Bub and H3K4me1 promote each other (Extended Figure.4e).

H4K16ac is not promoted by ATX nor H3K4me1

We examined H4K16ac in *atx1/2/r7* in two replicates. Only a few hypo-acetylated peaks were detected, which did not overlap with ATX1/2 bound genes nor ATX1/2/R7-marked genes (Extended Fig. 4e), suggesting that ATX are not the cause for H4K16ac. Testing if H4K16ac promotes ATX is currently difficult because no mutants that specifically alter H4K16ac have been reported as far as we know.

H3K36me3 is promoted by H3K4me1

This has been argued in the original manuscript. We strengthened the point by additional replicates of H3K36me3 in *atx12r7* and *ashh* mutants, as referred above.

Cis motifs are hard to test

For cis-motif the directionality is obvious. Proving causality, in this case, is technically difficult for the following reasons;

- Lack of DNA-binding domain of ATX protein suggests ATX-DNA interaction to involve other proteins, therefore Y1H or recombinant biochemistry is out of the choice.
- Another option to test causality is *in planta* engineering of the cis motifs. However, we think that testing the necessity of each cis motif this way will take a long time and effort and is out of the scope of this paper.
- Another option is genetically deleting Transcription Factors(TF) that are predicted to bridge ATX to motifs, such as Trihelix, TCP, and BP6. However, the expected redundancy of these TF families requires a lot of crossing and genotyping; ATX-tag lines are homo of two alleles (exogenous and endogenous ATX), therefore experiments with some TF's double mutant should be a quadruple homozygote.

As a piece of supporting evidence that the cis-motifs have some functions, we found that the highly weighted motifs, especially RGCCAW, are evolutionally conserved amongst the TSS region of land plants (Supplementary Figure.7).

Preceding researches support RNAP2-> H3K4me

Body of preceding research supports the view that SET1-type methyltransferase, to which ATXR3,7 belong, is subordinate to RNAP2.

- In yeast, where only one SET1-type methyltransferase is responsible for all H3K4me, transcriptional change is followed by H3K4me3 change, not the other way around (Kuang et al. 2014). The extent of H3K4 methylation is sensitive to genetic engineering of transcription rate (Soares et al. 2017).
- The molecular link between SET1 and RNAP2 is also elucidated in yeast. Polymerase II Associated Factor 1 Complex (PAF1C) is required for H3K4me, loss of PAF1C components depletes H3K4me in yeast. PAF1 interacts with RNAP2 and COMPASS, thus providing a scaffold for H3K4 methyltransferases to work with RNAP2 (Krogan et al. 2003)

Also in *Arabidopsis*, there are some indications that SET1-types are associated with RNAP2.

- IP-MS of S2Lb(a COMPASS component, homolog of Swd2) in *Arabidopsis* co-purified two proteins; ATXR3 and CDK9 homolog, which belong to P-TEFb that modulates elongation of RNAP2, suggesting a physical link, albeit indirectly, between RNAP2 and ATXR3 (Fiorucci et al. 2019).
- Mutation of PAF1C components and ATXR7 results in the same phenotype, such as reduced seed dormancy (Liu et al. 2011) and early flowering (He, Doyle, and Amasino 2004; Tamada, Yun, and Amasino 2009).

Finally, we examined genome-wide localization of RNAP2 in *atx12r7* and found that RNAP2 is *increased* over ATXR7-bound genes, contrary to the hypothesis that ATXR7 recruits RNAP2 onto chromatin. We plan to explore this phenomenon in our next study.

We believe these additional experimental validations better clarified the causative, especially for the H2Bub -> ATX pathway. However, considering the experimental limitations on some other pathways, we toned down the following sentences to avoid overstatement.

- The title of Fig.8, (Epi)genome-informed and cotranscriptional modes of H3K4 methylation -> Suggested models of (epi)genome-*

- These results suggest that SET1A is a cotranscriptional H3K4 methyltransferase, -> These results suggest that SET1A might be a cotranscriptional H3K4 methyltransferase,

Finally, in the SVM motif analysis, since ATXR7 binds to the TTS, it is not surprising that TSS motifs are not good predictor. The author should also consider testing with TTS 6-mers.

The ATXR7 models are built on TTS 6-mers. We are sorry for the poor explanation and thank you for pointing it out. We clarified the description.

Minor issues:

Introduction Line 37-41: needs elaboration and clarification.

We rewrote the corresponding section as follows;

Our previous genetic and genomic studies in *Arabidopsis thaliana* (hereafter *Arabidopsis*) have demonstrated key roles of the gene body H3K4me1 in various epigenetic phenomena. A putative H3K4me1 demethylase, LYSINE-SPECIFIC DEMETHYLASE1-LIKE2 (LDL2), mediates silencing of genes with a repressive histone mark H3K9me2 by removal of H3K4me1¹³. Another related demethylase, FLOWERING LOCUS D (FLD), downregulates transcriptional elongation and initiation by removing H3K4me1 in genes that have high levels of convergent overlapping transcription (i.e., antisense transcription)¹⁴.

Line 54 “methylase mutants with decreased H3K4me1 levels have not been reported” and line 58 “by increasing H3K4me1/2/3 level within the FLC locus” are conflicting. Please clarify.

Thank you for pointing this out. The pioneering study referred to in line 58 focused on very few loci, thus unable to differentiate major impacts from some secondary effects. Now the line 54 is “methylase mutants with a genome-wide decrease of H3K4me1 levels have not been reported”

Supplemental figure 2: the plots at the bottom needs better marking and explanation in legend.

Thank you. The legend was added.

It might be more conventional and easier to read to change “methylase” to “methyltransferase”

All “methylase” were replaced with “methyltransferase”.

The clustering in Fig. 1B has no scale.

Thank you for pointing it out. The scale was added.

Extended Fig 1b pattern agrees with the clustering in fig 1b very well and support the logic of making *atx1/2* double mutant. Overall I feel extended Fig 1b is a more informative and powerful figure to include than figure 1b.

Thank you. The figures were swapped accordingly.

Figure 1f: I don't understand why there is only one expected number of intersections, not three for the three possible pairwise comparisons?

That was because all ** -marked genes are $n=3,000$. Numbers within circles were 3,000 minus intersections. We changed the notation to make it clearer.

Fig 4e and 4f are a little hard to read, highlighting different circles for GAGA, TATA and telobox would be helpful.

Thank you for pointing it out. We changed the color and line style hoping for a better visualization.

Figure 8 has no legend and the idea that ATXR7 is recruited through COMPASS/PAF1c should be explained.

We added the legend, in which PAF1C is explained.

Reviewer #2 (Remarks to the Author):

This study identified ATXR1/2/r7 as H3K4me1 methylases. They compared H3K4me1 ChIP data in *atx* signal and triple mutants, and found that *atx 1/2/r7* triple mutants have dramatic decrease in H3K4me1. In addition, the authors analyzed ATXR1, 2, r7 ChIP data using random forest algorithm and found that ATXR7 is correlated with RNAP2, while ATXR1 and 2 are associated with other chromatin modifications. In general, this study demonstrated a role of ATXR1/2/R7 in accumulating H3K4me1. However, some data seems not consistent with each other, and several conclusion were not well supported by the data. I have several concerns:

We thank the Reviewer #2 for the positive evaluation and raising several critical points. We addressed these important points below.

1. Results shown in figure 1c and figure 1e are not consistent. In figure 1c *atx1/2/r7* had much stronger signal than *atx1/2*, however, in figure 1e, the H3K4me1 accumulation signals are similar in *atx1/2/r7* and *atx1/2*. How do you explain this difference?

Although it might look similar to the eyes, when the western blot H3K4me1 signal in Figure.1e was quantified and normalized to H3 signal (loading control), *atx12r7* shows a more prominent decrease than *atx1/2* (see the figure below), consistent with Figure.1c.

To further confirm *atx1*, *atx2*, *atxr7* having an additional impact for H3K4me1, we compared H3K4me1 levels in mutants with additional methods in this revision;

1. We quantified the differentially modified regions using a statistical model (MACS2);
 - We counted differentially H3K4 methylated regions by comparing mutants to WT. As expected, the numbers of the regions were *atx3*, *atx4*, *atx5* < *atx1*, *atx2*, *atxr7* << *atx12*, *atx1r7*, *atx2r7* << *atx12r7* (Supplementary Table.1). We also performed ChIP-seq replicates for {H3, H3K4me1} x {WT, *atx1*, *atx2*, *atxr7*, *atx1/2*, *atx1/r7*, *atx2/r7*, *atx1/2/r7*}, which confirmed the above trends (Supplementary Table.1 '*atx1,2,atxr7* replicate').
 - As a control, we also analyzed H3 occupancy similarly to H3K4me to show that rarely any regions are differentially occupied, validating the consistency among chromatin samples (Supplementary Table.1).
2. We utilized spike-in ChIP-seq for the above-mentioned replicates. Briefly, the arabidopsis chromatin was mixed with 'spike-in' chromatin of other organisms (in our case, *S.pombe*), then immunoprecipitated with H3K4me1 and H3 antibodies. *S.pombe* serves as an internal control. If H3K4me1 is decreased in a mutant, the % of *A.thaliana* reads normalized to H3 will decrease. Consistently, the % of *A.thaliana* reads normalized to H3 decreased roughly in the order of *atx1*, *atx2*, *atxr7* single mutants < double mutants < triple mutant (Extended Fig 1.c) with the exception of *atx1/r7*, possibly reflecting experimental noise.

2. Figure 2b, the researchers showed that ATX1 and ATX2 are preferentially located in the 5' end of gene, while ATX7 prefers to 3' end of gene. I suggest them to examine whether the H3K4me1 is decreased at the 3' end in *atx1/2* double mutant, but is decreased at the 5' end in *atxr7* single mutant.

Please correct us if we are wrong; we assume that you intended to ask whether H3K4me1 is decreased at the 5' end (TSS side of the genes) in *atx1/2* and 3' end in *atxr7*, consistent with the location of ATX(R) proteins. Indeed, H3K4me1 decreases at the 3' in the *atxr7* mutant, while the decrease is biased to 5' in *atx1/2* compared to *atxr7* (Fig. 1b,c).

On closer inspection, you might wonder how the decrease in ATX1/2 is not exactly at TSS, where ATX1/2 peaks. The rationale is that H3K4me1 is barely present in the TSS region (H3K4me1 covers the 3' gene body), so there is nothing to decrease in TSS. We interpret that ATX1/2 are recruited to and stalled at TSS, then exhibit enzymatic activity downstream. (In many cases, occupancy of enzymes does not match where its loss affects, which can be explained by several scenarios, such as the catalytic activities require cofactors, enzymes have active/inactive form which cannot be distinguished by FLAG-tag ChIP-seq, and redundant enzymes masking some of its effects.)

3. An important and unanswered question about H3K4me1 is whether it is the reason or the result of transcription. In figure 5A and B, the author plotted H3K4me1 signal vs gene expression in WT, *atx1/2*, and *atxr7*. They concluded that the positive correlation between transcription and H3K4me1 was disturbed by *atxr7*. However, a more interesting analysis could be that whether the genome-wide loss of H3K4me1 are associated with the change of gene expression in *atxr1/2*, *atxr7* or *atxr1/2/7*. For example, if there is a gene that is actively expressed and is enriched with H3K4me1 in WT, when the H3K4me1 in this gene is lost in *atx1/2/r7* mutants, is this gene expression lost or unchanged in *atx1/2/r7* mutants?

It should not be the choice between cause and effect, since they are non-exclusive, but better framed as two independent questions;

1. whether H3K4me1 results from transcription (Both supportive and negative studies exist.)
2. whether H3K4me1 activates transcription (it has been speculated, but lacks universal evidence or molecular mechanisms, as reviewed by Howe et al., 2017)

This work focuses on the first question by finding the rules behind H3K4methylase's chromatin binding (Figure. 3 and 4) and hypothesized that ATXR7-catalyzed H3K4me1 is induced by, and ATX1/2-catalyzed H3K4me1 are independent of transcription. Figure. 5 serve as a validation of this hypothesis. Let us clarify the logic behind Figure.5; the hypothesis predicts ATXR7 makes a correlation between H3K4me1 and transcription because ATXR7 deposits H3K4me1 on transcribed genes. Therefore, the loss of *ATXR7* will reduce the correlation between transcription and H3K4me1. On the other hand, *ATX1* and *ATX2* assumedly disturb this correlation by depositing H3K4me1 irrespective of transcription. Consequently, loss of *ATX1* and *ATX2* will 'unmask' the correlation between H3K4me1 and transcription. Consistent with these predictions, the *atxr7* mutant exhibits reduced correlation, while the *atx1/2* mutant exhibits increased correlation, providing support for the main hypothesis of this paper. Thus, Figure 5 is, although it might seem unfamiliar at first glance, a logically designed, necessary part of this paper.

In order to approach the second question, many previous works have adapted the analytic strategy that you suggested to see if the H3K4me change 'causes' transcriptional change, but the conclusion does not replicate amongst papers, failing to lead to a consensus. Our study here shows that this analytic strategy has a flaw: with the knowledge that some of the H3K4me results from transcription, when the loss of H3K4me overlaps with loss of transcription, that could be because a loss of transcription caused a loss of transcription-directed H3K4me by SET1, and not necessarily support the notion that 'H3K4me activates transcription'.

In any case, we have analyzed to find that genes that lost H3K4me1 in *atx12r7* significantly ($p\text{-val} \ll 0.01$) overlap with genes with reduced expression. We excluded this analysis from our paper because it is likely misinterpreted as supportive evidence for ‘H3K4me activates transcription’, and is irrelevant to the logic of the paper.

So, it remains a mystery, possibly the largest mystery in the field of euchromatin, what impacts (if any) H3K4me1 have for transcription. Several hypotheses in the field include the regulation of splicing and repression of cryptic transcription, and we also previously found that the presence of H3K4me1 overrides transcriptional repressor H3K9me2 (Inagaki et al., 2017). All in all, the role of H3K4me1 does not seem to be a simple quantity control of the steady-state amount of mRNA that can be quantified by mRNA-seq, but instead more intricate/indirect/conditional. The materials identified here (*atx12r7* and others) provide opportunities to test each hypothesis, and the insights we found (there are two types of H3K4me1) will allow us to conduct cautious analyses. Addressing that mystery will be a focus of our future study.

Minor concerns:

1: line 85, figure 1c, the author concluded that *atx1/atx2* double mutant had stronger effect than that in *atx1* and *atx2* single mutants. However, I think in figure 1c, the signals in *atx1* and *atx2* single mutants and *atx1/atx2* double mutant are quite similar.

Thank you for pointing that out. The figure format and color scale of Figure. 1c was optimized for better visualization.

- Fiorucci, Anne-Sophie, Clara Bourbousse, Lorenzo Concia, Martin Rougée, Gérald Zabulon, Elodie Layat, David Latrasse, et al. 2019. “Arabidopsis S2Lb Links AtCOMPASS-like and SDG2 Activity in H3K4me3 Independently from Histone H2B Monoubiquitination.” *Genome Biology*, 1–21.
- He, Yuehui, Mark R. Doyle, and Richard M. Amasino. 2004. “PAF1-Complex-Mediated Histone Methylation of FLOWERING LOCUS C Chromatin Is Required for the Vernalization-Responsive, Winter-Annual Habit in Arabidopsis.” *Genes & Development* 18 (22): 2774–84.
- Krogan, Nevan J., Jim Dover, Adam Wood, Jessica Schneider, Jonathan Heidt, Marry Ann Boateng, Kimberly Dean, et al. 2003. “The Paf1 Complex Is Required for Histone H3 Methylation by COMPASS and Dot1p: Linking Transcriptional Elongation to Histone Methylation.” *Molecular Cell* 11: 721–29.
- Kuang, Zheng, Ling Cai, Xuekui Zhang, Hongkai Ji, Benjamin P. Tu, and Jef D. Boeke. 2014. “High-Temporal-Resolution View of Transcription and Chromatin States across Distinct Metabolic States in Budding Yeast.” *Nature Structural & Molecular Biology* 21 (10): 854–63.
- Liu, Yongxiu, Regina Geyer, Martijn van Zanten, Annaick Carles, Yong Li, Anja Hörold, Steven van Nocker, and Wim J. J. Soppe. 2011. “Identification of the Arabidopsis REDUCED DORMANCY 2 Gene Uncovers a Role for the Polymerase Associated Factor 1 Complex in Seed Dormancy.” *PLoS One* 6 (7): e22241.
- Soares, Luis M., P. Cody He, Yujin Chun, Hyunsuk Suh, Tae Soo Kim, and Stephen Buratowski. 2017. “Determinants of Histone H3K4 Methylation Patterns.” *Molecular Cell* 68 (4): 773–85.e6.
- Tamada, Yosuke, Jae-Young Yun, and Richard M. Amasino. 2009. “ARABIDOPSIS TRITHORAX-RELATED7 Is Required for Methylation of Lysine 4 of Histone H3 and for Transcriptional Activation of FLOWERING LOCUS C.” *The Plant Cell* 21 (October): 3257–69.

- Howe, Françoise S., Harry Fischl, Struan C. Murray, and Jane Mellor. 2017. "Is H3K4me3 Instructive for Transcription Activation?" *BioEssays: News and Reviews in Molecular, Cellular and Developmental Biology* 39 (1): 1–12.
- Inagaki, Soichi, Mayumi Takahashi, Aoi Hosaka, Tasuku Ito, Atsushi Toyoda, Asao Fujiyama, Yoshiaki Tarutani, and Tetsuji Kakutani. 2017. "Gene-body Chromatin Modification Dynamics Mediate Epigenome Differentiation in Arabidopsis." *The EMBO Journal* 36 (8): e201694983.

REVIEWER COMMENTS

Reviewer #1 (Remarks to the Author):

Line 105 and Fig 1 f: it is odd that the targets are completely exclusive. For example, in *atx1/2*, H3K4me3 is affected on a set of genes but the H3K4me2 and H3K4me are unaffected (based on western blot). Who deposited these H3K4me2 and H3K4me on this set of genes?

I am still puzzled by the directionality of epigenome features and ATX1/2 binding. Line 169-172: Extended fig 3a does not clearly show that there is concomitant loss of H3K4me1 and H3K36me3, the r is quite modest. *Ashh2* mutant loses mostly 3' H3K36me3 but keeps the 5' H3K36me3, so maybe the 5' H3K36me3 in the *ashh2* mutant functions to maintain the H3K4me1, therefore it does not disprove that H3K36me3 drives ATX1/2 (which binds in TSS end anyway). The logic of line 173 – 174 on the ML to predict H3K36me3 is hard to follow and needs further explanation. Similarly the cause and effect of H2Bub and ATX1/2 is less clear because ATX1/2 binding is affected in *hub* mutant, but H2Bub is also lost in *atx1/2/7* mutant.

Minor:

Why is clustering of extend Fig 1b based on intragenic only?

Line 95- unclear- needs elaboration

Some figures are really hard to read.

Reviewer #3 (Remarks to the Author):

In this study, Oya et al., focused on the link between H3K4 mono-methylation, certain H3K4 methyltransferases and gene transcription (i.e., the transcriptional machinery). This manuscript is very interesting and supported by an incredibly large amount of data. I particularly appreciate the machine learning approach which seems to be a very effective tool. Previously, reviewers 1 and 2 already raised several important issues to which authors answered properly, making this manuscript all the more tangible and solid. From my side, I have only very few minor comments.

First, few references related to the function of H3K4 methylation were not cited and discussed:

- 1) Ding et al., 2011 (DOI: 10.1105/tpc.110.080150) described that ATX1 can bind the Ser5P of the RNAPII CTD while the random forest model describes a low importance for RNAPII Ser5P at ATX1-bound genes (Fig 3). Can this be related to a particular role of ATX1 for the transcriptional induction of certain abiotic stress related genes?
- 2) Liu et al., 2019 (10.1186/s13072-019-0285-6) demonstrated that H3K4me2 functions as a repressive mark in plants. This reference may appear at least in the introduction. This reference is particularly interesting in view of results described at lines 319-323.
- 3) Yanchao and Huang, 2018 (DOI: 10.1074/jbc.RA117.001390) demonstrated that SDG8 preferentially binds H3K4me1; Fiorucci et al., 2019 demonstrated that SDG2 activity is independent from H2Bub; both papers support finding described for example at line 426-427 (H2Bub promotes ATX1/2, while H3K36me3 is promoted by H3K4me3). These cross-talks are very interesting to me and rather complex even when considering few individual loci (e.g., Zhao et al., 2018 DOI : 10.1111/nph.15418)

Line 277-278: authors suggest the existence of mechanical links between sppRNA and ATX1/2. However, the location of sppRNAs coincides with promoter-proximal RNAPII stalling (usually unphosphorylated; DOI: 10.1038/s41467-020-16390-7) while no association was reported between ATX1/2 and RNAPII total CTD, Ser2P and Ser5P. In addition, I also regret that unphospho RNAPII data

(Zhu et al, 2018; 10.1038/s41477-018-0280-0) were not used for the random forest model or for Ext Fig 2. Related to this, the source of RNAPII data used is missing in Supp Table 4 (same comment for ATAC-seq).

In Fig 1e what is the difference between atx3/4/5 and atx3-1/4/5 ? The first does not present any global change in H3K4 methylation, while the second shows a decrease in H3K4me2 (i.e. and in H3K4me3 to a level comparable to atxr3).

Instead of showing atx3, 4 and 5, why not making appearing in figure 1b the heat map of the triple atx1/2/r7 as a metaplot?

Why not including H3K4me1, me2 and me3 in the random forest model as a proof of prediction validity?

I am wondering if the association between ATX1 and H3K27me3 visible in Fig. 3 as something to do with bistable/bivalent (H3K4me3/H3K27me3) chromatin?

(our response is in red)

Comments from reviewer 1

> *Line 105 and Fig 1 f: it is odd that the targets are completely exclusive. For example, in atxr3, H3K4me3 is affected on a set of genes but the H3K4me2 and H3K4me are unaffected (based on western blot). Who deposited these H3K4me2 and H3K4me on this set of genes?*

As the reviewer pointed out, the genes affected most by ATX1/2/R7 (to H3K4me1), ATX3/4/5 (to H3K4me2), and ATXR3 (to H3K4me3) are exclusive (Fig. 1f). These observations lead to the question; Who deposited H3K4me2 and H3K4me1 on genes in which ATXR3 deposits H3K4me3 (referred to as ATXR3-marked genes in this paper), for example? Our interpretation is that multiple methyltransferases from the ATX1/2/R7, ATX3/4/5, and ATXR3 groups (and/or other unidentified H3K4 methyltransferase(s)) jointly contribute to H3K4me1/2 on ATXR3-marked genes, therefore loss of any of the group does not reduce H3K4me1/2. In the previous manuscript we discussed this point at L406-408 in the Discussion section, which we revised in response to the reviewer's comment (L420-428 in the revised manuscript);

previous: The target genes of these three groups are mutually exclusive, and mutations within them affect H3K4me independently. For example, triple *atx1/2/r7* mutation causes the loss of only H3K4me1 and not H3K4me2/3. This observation implies that other ATX(R)s can catalyze H3K4me2/3 modifications on unmethylated H3K4 (H3K4me0) in their target regions. Accordingly, ATXR3 was shown to catalyze the H3K4me3 from H3K4me0 in vitro (Guo et al., 2010).

revised: The genes that are affected by the loss of these three groups are mutually exclusive, and mutations within them affect H3K4me independently. For example, triple *atx1/2/r7* mutation causes the loss of only H3K4me1 and not H3K4me2/3 on a set of genes (ATX1/2/R7-marked genes). This observation implies that other H3K4 methyltransferases can catalyze H3K4me2/3 modifications on unmethylated H3K4 (H3K4me0) in their target regions. Accordingly, ATXR3, for example, was shown to catalyze the H3K4me3 from H3K4me0 in vitro (Guo et al., 2010). In addition, the observation that H3K4me2/3 of ATX1/2/R7-marked genes being not largely affected by *atxr3* nor *atx3/4/5* mutation implies that multiple methyltransferases from the ATX1/2/R7, ATX3/4/5 and ATXR3 groups (and/or other unidentified H3K4 methyltransferase(s)) are redundantly involved in their deposition.

> *I am still puzzled by the directionality of epigenome features and ATX1/2 binding. Line 169-172: Extended fig 3a does not clearly show that there is concomitant loss of H3K4me1 and H3K36me3, the r is quite modest.*

We revised the correlation analysis of the corresponding figure (Supplementary Fig. 6a, which previously was Extended Fig.3a) calculating with fold change (log2 of mutant/WT) to calculating with the difference (mutant - WT). This better captures the linear relationship between H3K4me1 and H3K36me3 and visualizes the concomitant loss in the *atx1/2/r7* triple

mutants. The correlation estimate, based on the difference, is $\rho=0.66$. As another analytic approach, we visualized the H3K36me3 changes as heatmaps, sorted by H3K4me1 changes (Supplementary Fig.6a), which confirmed the concomitant decrease of H3K4me1 and H3K36me3. We also analyzed another ChIP-seq replicate to find the correlation is consistently positive ($\rho=0.51$, Supplementary Fig.6a).

> *Ashh2* mutant loses mostly 3' H3K36me3 but keeps the 5' H3K36me3, so maybe the 5' H3K36me3 in the *ashh2* mutant functions to maintain the H3K4me1, therefore it does not disprove that H3K36me3 drives ATX1/2 (which binds in TSS end anyway).

Indeed, the ideal genetic material for the test is a mutant that loses 5' H3K36me3 instead of 3', which the field lacks. We discussed that point as follows at L178-180 in the revised manuscript;

However, *ashh2* mutant keeps relatively high levels of H3K36me3 around TSS (Li et al. 2015) where ATX1/2 localize (Fig. 2b-d), thus we cannot exclude the possibility that H3K36me3 acts upstream in addition to downstream of ATX1/2-H3K4me1.

Additionally, we revised the analysis of *ashh2* to better demonstrate that H3K36me3 is weak in its ability to promote H3K4me1; Previous Extended Fig.3b, a kernel density plot of H3K4me1 reduction in *ashh2*, was replaced with a scatter plot and heatmaps visualizing the correlation (Supplementary Fig. 6b). The revised figure shows that changes of H3K36me3 levels in *ashh2* do not strongly correlate with changes of H3K4me1 ($\rho = -0.02$), in contrast to *atx1/2/r7* (Supplementary Fig. 6a, $\rho > 0.51$), in which hypo-H3K4me1 correlates with H3K36me3 loss.

> The logic of line 173 – 174 on the ML to predict H3K36me3 is hard to follow and needs further explanation.

We appreciate the reviewer pointing this out. We rewrote the corresponding lines (L180-184 in the revised manuscript);

previous: According to the random forest algorithm, the H3K36me3 reduction in *atx1/2/r7* was best explained by the combined H3K4me1 methylation activities of ATX1,2 and ATXR7 (Extended Fig. 3d), arguing against H3K36me being directly methylated by some or one of the ATX1,2 or ATXR7. These results suggest that H3K36me3 is regulated downstream of ATX1/2/R7-marked H3K4me1.

revised: To test the possibility that ATX1/2 proteins themselves direct H3K36me3 as well as H3K4me1, we trained another random forest model to predict H3K36me3 reduction in *atx1/2/r7* using H3K4me1 reduction in each and combined mutation(s) of *ATX1*, *ATX2* and *ATXR7*. The H3K36me3 reduction was best explained by the H3K4me1 reduction patterns of *atx1/2/r7* triple mutant, among others (Supplementary Fig. 6c), hinting that H3K4me1 itself rather than ATX1/2 act upstream of H3K36me3.

Also rewrote the legend of the corresponding figure (revised Supplementary Fig. 6c) ;

previous: Random forest was trained to predict H3K36me3-reduced region in *atx1/2/r7* based on H3K4me1 reduction in gene body regions of the *atx1*, *atx2*, *atxr7*, *atx1/2*, *atx1/r7*, *atx2/r7*, and *atx1/2/r7* mutants.

revised: If some of ATX1,2 and ATXR7 methylate H3K36, H3K36me3 reduction in *atx1/2/r7* mutant is expected to be biased towards the targets of the hypothetical multi-substrate methylase(s). In order to check for the bias, we sought to predict H3K36me3 reduction in *atx1/2/r7* by random forest, using per gene reductions of H3K4me1 in mutants made by all possible combination of *atx1*, *atx2* and *atxr7*, as explanatory features. H3K36me3 reduction was best explained by H3K4me1 reduction patterns in *atx1/2/r7* rather than biased towards some of the three methyltransferases.

> *Similarly the cause and effect of H2Bub and ATX1/2 is less clear because ATX1/2 binding is affected in hub mutant, but H2Bub is also lost in atx1/2/r7 mutant.*

As the reviewer pointed out, ATX1/2/R7-directed H3K4me1 and HUB-directed H2Bub promote each other. We made this point clear by revising the following lines (L197-215 in the revised manuscript);

previous: Conversely, H2Bub was decreased in H3K4me1-decreased genes in *atx1/2/r7* (Extended Fig. 4e), suggesting that H2Bub and H3K4me1 promote each other.

revised: Conversely, H2Bub was also decreased in H3K4me1-decreased genes in *atx1/2/r7* (Supplementary Fig. 7e). These results indicate that H2Bub and H3K4me1 are mutually promoting each other.

> *Minor:*

> *Why is clustering of extend Fig 1b based on intragenic only?*

H3K4me1 are exclusively found in genes and depleted from intergenic regions. For example, in our data, only 28 peaks of H3K4me1 (0.003%) out of 11035 peaks are intergenic. As our regions of interest are gene bodies, here we performed clustering based on intragenic only. In addition, including intergenic regions, which have quite low levels of H3K4me1, likely increase the noise of the analysis.

> Line 95- unclear- needs elaboration

We appreciate the reviewer pointing this out. We rewrote the corresponding lines (L96-98 in the revised manuscript) as follows:

previous: ChIP-seq analysis revealed hypo-H3K4me1 genes gain unmethylated H3K4 (H3K4me0) in *atx1/2/r7* (Supplementary Figure. 1b).

revised: ChIP-seq analysis targeting unmethylated H3K4 (H3K4me0) revealed that ATXR1/2/R7-marked genes showed increased signals of H3K4me0 in *atx1/2/r7* compared to wild type (Supplementary Fig. 2b).

> Some figures are really hard to read.

We revised the following figures, so as to make them easier to read.

- Enrichment/p-value legends of dotplots (Such as Supplementary Fig. 6d in the current manuscript) were revised to better represent continuous values.
- Text colors of Fig. 4ef, and dot plots are now white when the backgrounds are dark.
- We improved the resolution of several figures. Also we will reformat the current .jpeg figures to vector images in the final submission.

Comments from reviewer 3

Reviewer #3 (Remarks to the Author):

> In this study, Oya et al., focused on the link between H3K4 mono-methylation, certain H3K4 methyltransferases and gene transcription (i.e., the transcriptional machinery). This manuscript is very interesting and supported by an incredibly large amount of data. I particularly appreciate the machine learning approach which seems to be a very effective tool. Previously, reviewers 1 and 2 already raised several important issues to which authors answered properly, making this manuscript all the more tangible and solid. From my side, I have only very few minor comments.

First of all, we thank Reviewer #3 for their very positive evaluation of our manuscript and constructive comments, which we believe improved our revised manuscript.

> First, few references related to the function of H3K4 methylation were not cited and discussed:

> 1) Ding et al., 2011 (DOI: 10.1105/tpc.110.080150) described that ATX1 can bind the Ser5P of the RNAPII CTD while the random forest model describes a low importance for RNAPII Ser5P at ATX1-bound genes (Fig 3). Can this be related to a particular role of ATX1 for the transcriptional induction of certain abiotic stress related genes?

We agree that this literature is important. Indeed, Ding *et al.* demonstrated that ATX1 binds the RNAP2-Ser5P. While our results indicate that RNAP2 is not the major recruiter for ATX1/2, they do not contradict the ability of ATX1/2 to form a physical link with RNAP2. We clarified the relation between Ding *et al.*'s and our results by adding the following comments on L220-223;

A previous study reported that physical interaction between ATX1 and RNAP2 phospho-Ser5 is involved in the recruitment of ATX1 to several ATX1-regulated genes³⁵. Our results do not exclude the possibility that phospho-RNAP2 is also involved in the chromatin recruitment of ATX1/2 in addition to chromatin modification(s) such as H2Bub (Fig. 3, Supplementary Fig. 7)

In light of this comment, we noticed that the average RNAP2 profiles of Extended Fig.2 were misleading and revised them. Previously, the average profile represented the mean, a metric sensitive to outliers. ATX1/2-unbound genes have a higher mean than ATX1/2-bound genes, perhaps giving the false impression that ATX1/2 are exclusive to RNAP2. Other metrics

robust to outliers, such as median or mean of log values, indicate that ATX1/2 somewhat colocalizes with RNAP2. The revised Supplementary Fig. 4 represents the mean of the log(RNAP2).

It is also possible that, as the reviewer suggested, ATX1 works differently on abiotic stress genes. The role of ATX1 for conditional transcriptional inducers is a highly interesting possibility but we would like to refrain from too much speculation at this stage.

> 2) Liu et al., 2019 (10.1186/s13072-019-0285-6) demonstrated that H3K4me2 functions as a repressive mark in plants. This reference may appear at least in the introduction. This reference is particularly interesting in view of results described at lines 319-323.

We agree that this point is worth discussing. We added the following line in L106-110.

For example, ATX345-marked genes showed lower levels of expression compared to others (Supplementary Fig. 1e), in line with the body of research that suggests that H3K4me2 is a repressive mark in plants; H3K4me2 colocalizes with other repressive marks such as H3K27me3 (Roudier et al. 2011) and anti-correlates with transcription (Yuhao Liu et al. 2019). Hypo-H3K4me2 activates transcription in rice (Yuhao Liu et al. 2019) and during regeneration of Arabidopsis (Ishihara et al. 2019).

> 3) Yanchao and Huang, 2018 (DOI: 10.1074/jbc.RA117.001390) demonstrated that SDG8 preferentially binds H3K4me1; Fiorucci et al., 2019 demonstrated that SDG2 activity is independent from H2Bub; both papers support finding described for example at line 426-427 (H2Bub promotes ATX1/2, while H3K36me3 is promoted by H3K4me3). These cross-talks are very interesting to me and rather complex even when considering few individual loci (e.g., Zhao et al., 2018 DOI : 10.1111/nph.15418)

As the reviewer brought up, the view that H3K4me1 promotes H3K36me3 itself was consistent with Yanchao and Huang's findings. However, to our surprise, hypo-H3K36me3 regions in the *atx1/2/r7* mutant were not enriched with SDG8(ASHH2)-marked regions (Supplementary Fig. 6d, which previously was Extended Fig. 3e). This perhaps indicates a presence of another recruiting mechanism for ASHH2 overpowering H3K4me1-targeting mediated by the CW domain, or that the binding specificity of the CW domain of ASHH2 may be modified by other factors *in vivo*. We added the following line to L185-188.

previous: The *ashh3* mutant affects H3K36me3 at ATX1/2/R7-marked genes, consistent with the view that ASHH3 functions downstream of ATX1/2/R7-marked H3K4me1 to mediate H3K36me3 (Extended Fig. 3e, f).

revised: Among mutants for five H3K36 methyltransferase family genes, the *ashh3* mutant affects H3K36me3 at ATX1/2/R7-marked genes, while others including ASHH2, which has a H3K4me1-binding domain³⁴, marks genes that are not largely overlapped with ATX1/2/R7-marked genes (Supplementary Fig. 6d, e), consistent with the view that ASHH3 functions downstream of ATX1/2/R7-marked H3K4me1 to mediate H3K36me3.

To better contextualize Fiorucci et al. 2019 results, we revised the L450-452;

previous: The requirement of H2Bub for H3K4me modification has long been known, and various explanatory mechanisms have been proposed.

revised: H2Bub has long been known to promote H3K4 methyltransferase in yeasts and mammals and various explanatory mechanisms have been proposed, while in plants ATXR3 was found not to follow the rule (Fiorucci et al. 2019).

We appreciate the reviewer's suggestions very much. We believe the incorporation of these literatures improved the discussion in our manuscript very much.

> Line 277-278: authors suggest the existence of mechanical links between sppRNA and ATX1/2. However, the location of sppRNAs coincides with promoter-proximal RNAPII stalling (usually unphosphorylated; DOI: 10.1038/s41467-020-16390-7) while no association was reported between ATX1/2 and RNAPII total CTD, Ser2P and Ser5P.

We have three interpretations.

- RNAP2 stalling in the sppRNA regions is not prominent in our sample condition. Although the paper and our sample conditions are close (both MS plate LD 22°C 14 dai or 15 dai), the RNAP2 dynamics are highly sample-specific
- Unphos-RNAP2 is a highly specific recruiter, which is not included in our data as the reviewer pointed out.
- Actually, some association was observed between ATX1/2 localization and RNAP2 as shown below for the total RNAP2 level, and the association was stronger for sppRNA-harboring genes and sppRNA-harboring ATX1/2-bound genes, although the associations were much stronger between ATXR7 and RNAP2-Ser2P/Ser5P (Fig 3a,g; Supplementary Fig. 4). These results are consistent with that of Thomas et al. (2020) paper and our random forest results.

In any case, we now see 'mechanical link' as an overstatement, only supported by the paucity of evidence and perceived connection with MII3/4 regulating enhancer RNA. We toned down 'mechanical link' to 'some link.'

> In addition, I also regret that unphospho RNAPII data (Zhu et al, 2018; 10.1038/s41477-018-0280-0) were not used for the random forest model or for Ext Fig 2.

Related to this, the source of RNAPII data used is missing in Supp Table 4 (same comment for ATAC-seq).

We generated RNAP2 ChIP-seq and ATAC-seq data for this study, which we clarified by adding explanations in Supplementary Table 4 in response to this comment. While there are many interesting data sources that capture transcriptional dynamics, such as pNET-seq datasets the reviewer suggested, we reasoned it is better to use the in-house data, generated with the uniform conditions as the other datasets such as H3K4me and ATX(R)s ChIP-seq, because transcription related features (expression, RNAP2 localization, and open chromatin regions) can be variable and depend on growth condition/stage.

> In Fig 1e what is the difference between atx3/4/5 and atx3-1/4/5 ? The first does not present any global change in H3K4 methylation, while the second shows a decrease in H3K4me2 (i.e. and in H3K4me3 to a level comparable to atxr3).

atx3/4/5 triple mutant is made with *atx3-2* allele (just referred to as *atx3*), while *atx3-1/4/5* is with *atx3-1* allele. The former allele, previously used in Tamada 2009, has T-DNA inserted in 5' UTR and thus likely be a weak allele, while the *atx3-1* allele, characterized in Chen 2017, has exon insertion. Indeed, *atx3-1/4/5* shows a stronger reduction of H3K4me as shown as western blot as well as ChIP-seq.

We mainly analyzed *atx3/4/5* with weak allele *atx3-2* because this triple mutant has perceived stronger specificity for H3K4me2 than *atx3-1/4/5* and therefore it is likely useful for characterizing specific function(s) of H3K4me2. We mentioned the difference between *atx3-1* and *atx3-2* in the Materials and Methods section and refer to that in the relevant part of the main text (L331-332).

> Instead of showing atx3, 4 and 5, why not making appearing in figure 1b the heat map of the triple atx1/2/r7 as a metaplot?

We expect Figure.1b visually helps the reader to understand that we screened for H3K4me1-reducing mutant from the screening pool containing *atx3, atx4* and *atx5*, instead that we focused on *atx1, atx2* and *atxr7* from the start. Adding a metaplot of *atx1/2/r7* would make a message redundant to the heatmap in Fig. 1c and the metaplots in Supplementary Fig 1f. We would like to keep Figure.1b as it is.

> Why not including H3K4me1, me2 and me3 in the random forest model as a proof of prediction validity?

Levels of H3K4me1,2,3 are not the strongest predictors of ATX(R) localizations (figures below). This observation is in line with homologs of ATX(R)s; for example, MII3/4 localization anti-correlates with H3K4me1 on a subset of enhancers (Dorghi et al., 2017). We reason that enzymes and products can dislocate in multiple scenarios; enzymes can have catalytically inactive/poised states, catalytic products can be shared by multiple enzymes, and after reaction, the enzymes disassociate from chromatin while modifications last longer. We validated the random forest models with violin plots and experimental depletion of candidate recruiters, which we think are straightforward and appropriate validations of the models.

> I am wondering if the association between ATX1 and H3K27me3 visible in Fig. 3 has something to do with bistable/bivalent (H3K4me3/H3K27me3) chromatin?

We agree that is an interesting point. Additional observations such as ATX1 (and also ATX2) localize on Polycomb Repressive Elements (Fig. 4), and previous research describing ATX1's roles in stress and environmental responses and control of developmental genes also suggests some link between H3K27me3 and ATX1 (and perhaps ATX2).

Considering that the association is relatively weak, and the doubt on the interpretation that bivalency is a mark of a poised state (e.g., Shah et al., 2021, bioRxiv), we believe that further discussion requires a more cautious and rigorous analysis, which would require a separate paper.

REVIEWERS' COMMENTS

Reviewer #1 (Remarks to the Author):

The authors have addressed most of my questions in the previous round of revision. I only have the following minor points:

Check all figures and supplemental figures to ensure that all axis and legends are labeled. For example, in Figure 1C - label the heat map color key

Line 107: Figure s1e: should include a p-value to show that the lower expression level is significant

Line 180-184: the machine learning does not indicate the directionality of H3K4me1 act upstream of H3K36me3. I suggest removing this part to avoid over-interpretation by audience.

Reviewer #3 (Remarks to the Author):

Author did a really great job. I am very fine with the actual version, as well as with author's answers to reviewers comments. I was a real pleasure to review this manuscript !

Reviewer #1 (Remarks to the Author):

The authors have addressed most of my questions in the previous round of revision. I only have the following minor points:

Check all figures and supplemental figures to ensure that all axis and legends are labeled. For example, in Figure 1C - label the heat map color key

Thank you for pointing that out. We checked and made several improvements including Figure 1C.

Line 107: Figure s1e: should include a p-value to show that the lower expression level is significant.

The p-value of Dunnett's test (multiple comparisons of means) is now included to demonstrate the significant difference.

Line 180-184: the machine learning does not indicate the directionality of H3K4me1 act upstream of H3K36me3. I suggest removing this part to avoid over-interpretation by audience.

We agree that the lines have a risk of over-interpretation. We removed the corresponding lines and Supplementary Figure 6c.

Reviewer #3 (Remarks to the Author):

Author did a really great job. I am very fine with the actual version, as well as with author's answers to reviewers comments. I was a real pleasure to review this manuscript !

We have been delighted and honored by the careful and constructive attention of reviewers. The manuscript has improved greatly with your suggestions and comments. Thank you both very much for reviewing our manuscript.